# Mechanism of DNA unwinding by MCM8-9 in complex with HROB

Ananya Acharya[1,2], Hélène Bret [3], Jen-Wei Huang [4], Martin Mütze[5], Martin Göse[5], Vera Maria Kissling [2,6], Ralf Seidel [5], Alberto Ciccia [4], Raphaël Guérois [3] ✉ & Petr Cejka [1,2] ✉

HROB promotes the MCM8-9 helicase in DNA damage response. To understand how HROB activates MCM8-9, we defined their interaction interface. We showed that HROB makes important yet transient contacts with both MCM8 and MCM9, and binds the MCM8-9 heterodimer with the highest affinity. MCM8-9-HROB prefer branched DNA structures, and display low DNA unwinding processivity. MCM8-9 unwinds DNA as a hexamer that assembles from dimers on DNA in the presence of ATP. The hexamer involves two repeating protein-protein interfaces between the alternating MCM8 and MCM9 subunits. One of these interfaces is quite stable and forms an obligate heterodimer across which HROB binds. The other interface is labile and mediates hexamer assembly, independently of HROB. The ATPase site formed at the labile interface contributes disproportionally more to DNA unwinding than that at the stable interface. Here, we show that HROB promotes DNA unwinding downstream of MCM8-9 loading and ring formation on ssDNA.

DNA helicases are motor proteins that move directionally along a nucleic acid phosphodiester backbone, separating two strands of a DNA double-helix. The minichromosome maintenance (MCM) proteins are a subfamily of hexameric DNA helicases belonging to the AAA+ ATPase family that function in diverse cellular processes, including DNA replication and repair[1,2]. In humans, the MCM helicase family contains eight members (MCM2-9)[3]. The best characterized are the MCM2-7 proteins that form the motor of the CMG (Cdc45-MCM-GINS) DNA replicative helicase. Recent advances using biochemical and structural approaches uncovered that MCM2-7 is recruited as a single hexameric open ring together with Cdt1 by the Origin Recognition Complex (ORC) and Cdc6 to dsDNA, followed by another MCM2-7-Cdt1 complex to form an inactive double hexamer[4–6]. Subsequently, the MCM2-7 complexes are turned into two active single hexameric CMG helicases encircling single-stranded DNA (ssDNA), where MCM2-7 is stimulated by GINS and Cdc45[5,7,8]. The loading and activation of

MCM2-7 is quite complex, likely reflecting the necessity to restrict the initiation of DNA replication to take place only once during the cell cycle at the onset of S-phase.

Other AAA+ helicases, including archaeal replicative MCM proteins, also form hexamers. However, they assemble from only one polypeptide. Archaeal MCM helicases similarly form double-hexamers prior to activation, however, as all other AAA+ helicases, function as single hexameric rings during DNA unwinding[9]. The loading mechanism and activation of the MCM and other AAA+ helicases may differ, yet several characteristics are common to most members of this family characterized to date[1,2]. These proteins contain an N-terminal oligonucleotide-oligosaccharide-binding (OB-fold) domain, which is involved in protein oligomerization and DNA binding. The conserved C-terminal parts of the protein harbor the AAA+ ATPase. Characteristically, a functional ATPase site is reconstituted from two adjacent subunits: the Walker A and B motifs are provided by one polypeptide,

[1]Institute for Research in Biomedicine, Università della Svizzera italiana (USI), Faculty of Biomedical Sciences, Bellinzona 6500, Switzerland. [2]Department of Biology, Institute of Biochemistry, Eidgenössische Technische Hochschule (ETH), Zürich 8093, Switzerland. [3]Université Paris-Saclay, CEA, CNRS, Institute for Integrative Biology of the Cell (I2BC), 91198 Gif-sur-Yvette, France. [4]Department of Genetics and Development, Institute for Cancer Genetics, Herbert Irving Comprehensive Cancer Center, Columbia University Irving Medical Center, New York, NY, USA. [5]Peter Debye Institute for Soft Matter Physics, Universität Leipzig, Leipzig 04103, Germany. [6]Present address: Particles-Biology Interactions Laboratory, Department of Materials Meet Life, Swiss Federal Laboratories for Materials Science and Technology (Empa), St. Gallen 9014, Switzerland. ✉e-mail: raphael.guerois@cea.fr; petr.cejka@irb.usi.ch

while another component, termed the arginine finger, belongs to the neighboring subunit[10–12]. Consequently, ATP is hydrolyzed at the interface of the two subunits, typically in a sequential manner along the ring structure, and hexamer formation is hence a prerequisite for DNA unwinding activity. While the MCM complexes may exist as planar rings, during DNA unwinding, they form a spiral staircase-like structure that employs a "hand-over-hand" mechanism to move along ssDNA[1,2].

The MCM8 and MCM9 proteins are only found in a subset of multicellular eukaryotes[3,13]. They are missing in fungi and nematodes, but present in most plants and vertebrates, while *Drosophila* only contains an MCM8 homologue[14–16]. The MCM8 and MCM9 proteins associate together (MCM8-9) and are similarly thought to form hexamers[17–20], yet their assembly, function and mode of activation remain poorly understood. While MCM8 has the standard domain structure, MCM9 contains an unusual C-terminal extension downstream of the AAA+ ATPase site[21]. In humans, defects associated with MCM8-9 were linked to primary ovarian failure and hence infertility[22,23], and defects or overexpression of MCM8-9 may promote cancer[24–27]. Most reports to date suggest that MCM8-9 functions in meiosis and in DNA repair, particularly in homologous recombination (HR) in response to DNA interstrand crosslinks (ICLs), as well as to maintain replication fork stability[15–17,21,28–30]. Mutant cell lines show only minor sensitivity to ionizing radiation or bleomycin, suggesting that MCM8-9 is not a universal DNA break repair factor[31,32]. Rather, MCM8-9 may function in recombination, assisting ICL repair in the context of stalled replication forks, possibly alongside Fanconi anemia proteins[31–33]. MCM8-9 was also proposed to support recombination-based DNA synthesis to allow the extension of the invading DNA strand[31,32]. However, MCM8-9 was also suggested to function at the onset of recombination during DNA end resection in conjunction with the MRN complex[34] or to help facilitate RAD51 loading[35]. Therefore, there are reports that implicate MCM8-9 to act both upstream and downstream of RAD51 during HR. Beyond recombination, it was proposed that MCM8-9 may unwind DNA during mismatch repair[36], an observation that was supported by the identification of mutant alleles associated with the Lynch syndrome or microsatellite instability[29,37,38]. Finally, while MCM8-9 is not required for general DNA replication[39], it may have a residual function in DNA synthesis in the absence of MCM2-7, particularly on damaged DNA templates[40].

Recently, several groups identified a protein named HROB (homologous recombination factor with OB-fold, also termed MCM8IP or C17orf53)[31,33,41]. Defects in HROB cause pronounced meiotic defects, ICL sensitivity, and recombination impairment that resembled and are epistatic with defects in MCM8-9[31,33,41]. RAD51 loading in HROB-deficient cells was normal, however, leading to persistent RAD51 foci, hinting that HROB with MCM8-9 might act at the postsynaptic HR stage downstream of RAD51[31]. While HROB has no apparent catalytic activity, it was found to physically interact with MCM8-9, and to stimulate its DNA unwinding activity[33]. The siRNA-mediated depletion of HROB compromised the accumulation of MCM8 at DNA repair foci but not vice versa, suggesting that HROB acts upstream of MCM8-9[31]. It was inferred that HROB may help load MCM8-9 on DNA, yet the mechanism of MCM8-9 loading on DNA and activation was not demonstrated[31,33,41].

Using molecular modeling, ensemble, and single-molecule biochemistry, we define here the physical and functional interactions between HROB and MCM8-9. HROB interacts with both MCM8 and MCM9 subunits. The OB-fold domain of HROB does not support DNA binding, but is essential for its interaction with MCM9 and hence the stimulation of the MCM8-9 helicase activity. MCM8-9 in conjunction with HROB prefers to bind and unwind branched DNA structures, and single-molecule experiments with magnetic tweezers revealed that DNA unwinding by the ensemble is not processive. We show that MCM8-9 forms hexamers that assemble from dimers on DNA, and ATP helps to lock the MCM8-9 ring on single-stranded DNA (ssDNA).

Unexpectedly, HROB does not affect the oligomerization, loading, or closing of the MCM8-9 ring on ssDNA, and it does not affect its substrate preference. Rather, the reconstituted in vitro assays demonstrate that HROB primarily promotes MCM8-9 downstream of its assembly on DNA to stimulate specifically translocation and productive DNA unwinding.

## Results

### HROB interacts with both subunits of the MCM8-9 heterodimer

Previously, HROB was found in a complex with MCM8-9[31,33,41]. To better characterize their interaction, we have expressed and purified recombinant MCM8-9 as a complex, as well as the HROB protein. The MCM9 protein was fused with an MBP-tag at the N-terminus, while MCM8 and HROB contained an N-terminal and C-terminal FLAG-tags, respectively, to facilitate expression and purification[33]. We noted that the presence of the MBP tag did not affect DNA unwinding by MCM8-9 with HROB (Supplementary Fig. 1a) using established conditions[33]. As MCM8-9 was unstable during purification and upon tag cleavage, the MBP tag was retained for most experiments unless noted otherwise. We next employed mass photometry to monitor the interaction between MCM8-9 and HROB. While HROB was largely monomeric, the major species in the recombinant MCM8-9 preparation corresponded to a heterodimer (other oligomeric species will be described later in the text) (Fig. 1a, b). A combination of equimolar concentrations of MCM8-9 and HROB yielded a species with a molecular weight corresponding to the MCM8-9·HROB complex, i.e. one HROB molecule bound to one MCM8-9 heterodimer (Fig. 1c). However, we note that the interaction was unstable, as the complex largely fell apart upon dilution (Supplementary Fig. 1b). Likewise, in pulldown experiments with immobilized HROB, we obtained substoichiometric amounts of MCM8-9 (Supplementary Fig. 1c, d).

Previous experiments with cell extracts showed that HROB formed a complex with MCM8-9 via a region spanning residues 396 to 413 of HROB, but also via residues in HROB from 432 to the end of the protein[33]. The primary structure of HROB contains a proline-rich region (PRR18) with an unknown function, and an oligonucleotide-oligosaccharide-binding (OB-fold) domain (Fig. 1d). Overall, HROB is predicted to be largely unstructured, containing 79% disordered regions according to MobiDB[42], with the exception of the OB-fold domain. OB-folds often mediate protein-DNA and/or protein-protein interactions, and the role of the OB-fold in HROB remained unknown. To determine its function, we expressed and purified an internally truncated HROB variant lacking the OB-fold domain spanning residues 492-575, HROB-ΔOB (Fig. 1d, e). HROB-ΔOB proficiently bound ssDNA comparably to the wild type protein (Fig. 1f and Supplementary Fig. 1e, f), but was entirely deficient in promoting the helicase activity of MCM8-9 (Fig. 1g, h). These data show that during DNA unwinding, the OB-fold domain of HROB is not likely to be involved in DNA binding and may be primarily responsible for the interaction with MCM8-9 (Supplementary Fig. 1g), representing a yet uncharacterized interaction interface.

We next set out to define the physical interaction between HROB and MCM8-9. AlphaFold2 predicted that MCM8-9 forms a hexamer (Fig. 1i), which is very similar to the recent cryoEM structure obtained with human MCM8-9 N-terminal domains[19], chicken MCM8-9[20], or truncated human MCM8-9 complex[18]. The structural model predicts one molecule of HROB to bind both subunits of the MCM8-9 heterodimer (Fig. 1i), in agreement with our mass photometry experiment (Fig. 1c). The model indicated a possible interaction between the OB-fold domain of HROB (446-580) and the hinge between the N-and C-terminal domains of MCM9, located on the outer side of the MCM8-9 hexameric complex (Fig. 1i, j). Next, upstream of the HROB OB-fold, an unstructured region spanning residues 362-440 was predicted to bind MCM8 over an extensive surface located at the hinge between the N- and C-terminal domains

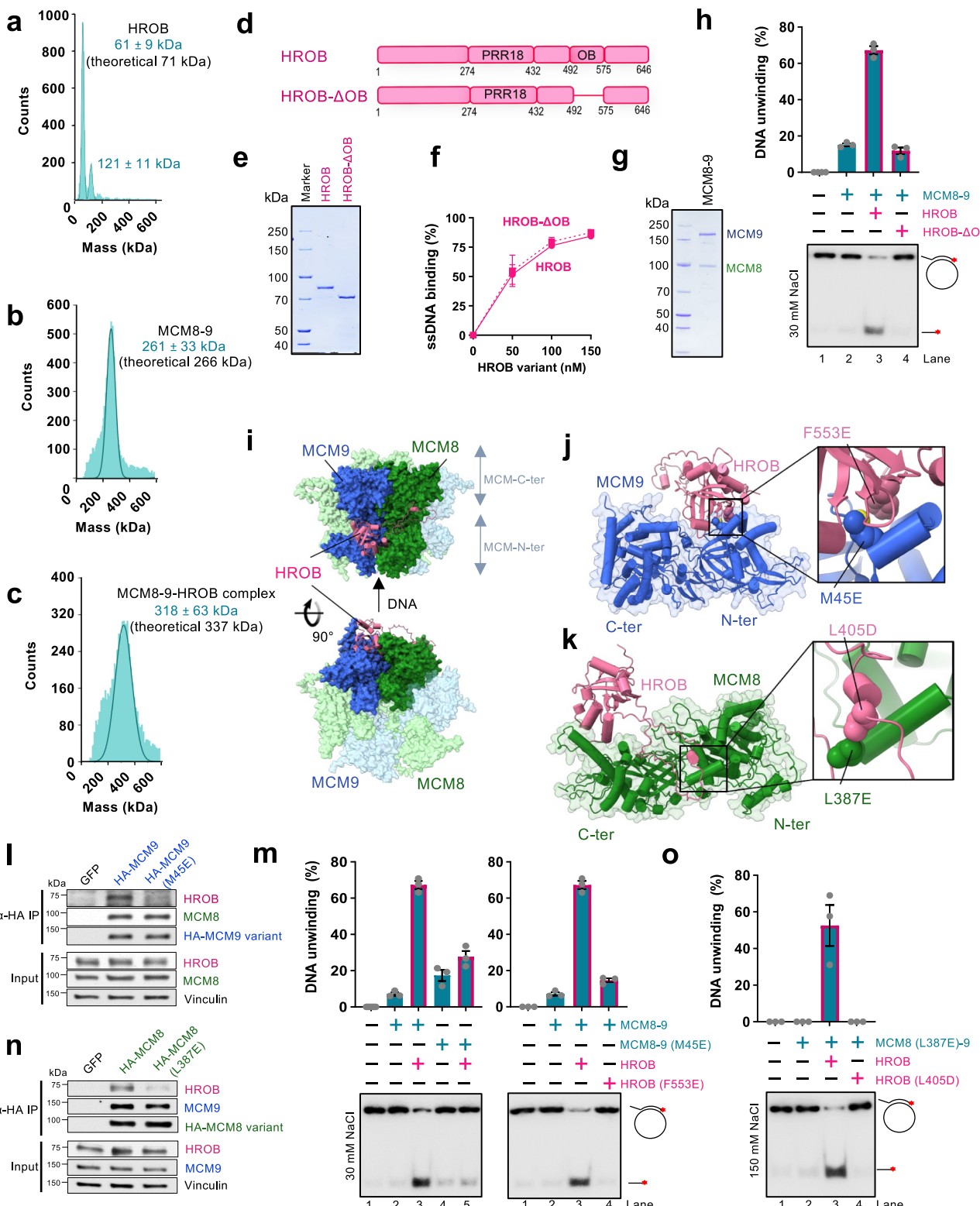

of the helicase (Fig. 1i,k), in agreement with the fragment analysis reported previously[33].

To validate the predicted structures, we designed several mutations at positions located in the interface between HROB and MCM8, as well as between HROB and MCM9 (Fig. 1j,k). Two conserved and apolar residues, HROB-F553 and MCM9-M45, anchored in the interface between the OB-fold of HROB and the N-terminal domain of MCM9, were substituted into a glutamate to destabilize their assembly (Fig. 1j and Supplementary Fig. 1h). We observed that

the single point mutations either on the MCM9 side (M45E), or within the OB-fold of HROB (F553E) strongly reduced physical interactions as assayed with recombinant proteins in vitro or in cell extracts (Fig. 1l and Supplementary Fig. 2a,b). Consequently, both single-point mutants were strongly impaired in DNA unwinding, confirming the functional relevance of the interaction interface between MCM9 and HROB (Fig. 1m). To validate and assess the importance of the predicted interaction of HROB with the MCM8 subunit, we focused on the most conserved positions in the

**Fig. 1 | HROB interacts with both subunits of the MCM8-9 heterodimer.**
**a** Measured molecular weight distribution of FLAG-tagged HROB by mass photo-
metry. Error, SD. **b** Measured molecular weight distribution of MCM8-9 (FLAG-
tagged MCM8 and MBP-tagged MCM9) by mass photometry. Error, SD. **c** Measured
molecular weight distribution of MCM8-9-HROB co-complex (1:1 complex of
MCM8-9 dimer:HROB) by mass photometry. Error, SD. **d** Schematic of HROB and
HROB-ΔOB. **e** Representative gel showing purified FLAG-HROB and FLAG-HROB-
ΔOB, $n = 2$ independent experiments. **f** Quantification of assays such as shown in
Supplementary Fig. 1e. Error bars, SEM; $n = 3$ independent experiments. **g** Repre-
sentative gel showing purified MCM8-9, $n = 2$ independent experiments. **h** DNA
unwinding of M13-based circular DNA by MCM8-9 (100 nM) and HROB or HROB-
ΔOB (50 nM) at 30 mM NaCl. Red asterisk, radioactive label. Top, quantitation;
error bars, SEM; bottom, representative of $n = 3$ independent experiments.
**i** Schematic of HROB binding to MCM8-9 hexamer modeled using AlphaFold2.
**j** Schematic of the interaction model between MCM9 and HROB. Interacting resi-
dues HROB-F553 and MCM9-M45 are highlighted. F553E and M45E mutations were
further tested. **k** Schematic of the interaction model between MCM8 and HROB.

Interacting residues HROB-L405 and MCM8-L387 are highlighted. The L405D and
L387E mutations were further tested. **l** MCM9-M45E disrupts MCM9-HROB inter-
action. Lysates expressing GFP, HA-MCM9-WT or HA-MCM9-M45E were subjected
to HA-immunoprecipitation. Immunoblotting of HROB, MCM8, and HA is pre-
sented, $n = 2$ independent experiments. **m** DNA unwinding of M13-based circular
DNA with indicated mutants (100 nM) to test the impact of disrupted HROB
(50 nM) interaction with MCM9 at 30 mM NaCl. Wild type HROB with wild type
MCM8-9 is replotted as in panel h for reference. Red asterisk, radioactive label. Top,
quantitation; error bars, SEM; bottom, representative of $n = 3$ independent
experiments. **n** MCM8-L387E disrupts MCM8-HROB interaction. Lysates expressing
GFP, HA-MCM8-WT, and HA-MCM8-L387E were subject to HA-
immunoprecipitation. Immunoblotting of HROB, MCM9, and HA is presented, $n = 2$
independent experiments. **o** DNA unwinding assay as in panel m for disrupted
HROB (25 nM) interaction with MCM8 (50 nM) at 150 mM NaCl. Red asterisk,
radioactive label. Top, quantitation; error bars, SEM; bottom, representative of
$n = 3$ independent experiments.

disordered tail of HROB and analyzed their contacts with MCM8
residues. HROB-L405 was modelled to form an apolar contact with
MCM8-L387 (Fig. 1k). This contact is predicted by AlphaFold2 to be
conserved in non-vertebrate and plant species (Supplementary
Fig. 2c,d). Probing for this second interacting region, mutating
individually L387E in MCM8 or L405D in HROB reduced the physical
interaction in cell extracts (Fig. 1n) and in pulldowns with recombi-
nant proteins (Supplementary Fig. 2a,b). In DNA unwinding, the
individual mutations of the HROB-MCM8 interface initially did
not cause notable defects (Supplementary Fig. 2e,f). However,
when MCM8-L387E and HROB-L405D mutations were combined
and more restrictive reaction conditions were employed, we
observed a notable impairment of DNA unwinding (Fig. 1o). Toge-
ther, we establish that there are at least two functionally important
interfaces between HROB and MCM8-9, one on each subunit of
the complex.

## Inhibition of MCM8-9 by CDK phosphorylation
Cyclin-dependent kinase (CDK) regulates the activity of MCM2-7
with respect to the cell cycle stage on multiple levels[43,44]. We have
noted in human cell extracts that the MCM9 subunit exhibited
changes in electrophoretic mobility upon treatment with λ-phos-
phatase, indicating that MCM8-9 may be also subject to phosphor-
ylation (Supplementary Fig. 3a). Interestingly, the C-terminal
extension of MCM9 contains a large number of putative CDK
phosphorylation sites with unknown function (Supplementary
Fig. 3b). To test whether phosphorylation of MCM8-9 regulates its
capacity to unwind DNA in vitro, we prepared the recombinant
MCM8-9 complex in insect cells without or with phosphatase inhi-
bitors, and treated the complex with λ-phosphatase (the one pre-
pared without phosphatase inhibitors) during purification
(Supplementary Fig. 3c). We noted that recombinant MCM9 exhi-
bited changes in its electrophoretic mobility (Supplementary
Fig. 3c), resembling what we found in human cell extracts (Supplementary
Fig. 3a). DNA binding and unwinding by MCM8-9 was generally
inhibited by salt (Supplementary Fig. 3d,e). We observed that under
physiological ionic strength conditions (150 mM salt), phosphatase
treatment of phosphorylated MCM8-9 stimulated DNA unwinding in
the presence of HROB (Supplementary Fig. 3f), in agreement with
the comparison of non-phosphorylated and phosphorylated MCM8-
9 variants expressed in insect cells, in assays at 150 mM NaCl (Sup-
plementary Fig. 3d). The moderate inhibition of the MCM8-9 com-
plex upon phosphorylation could be recapitulated upon treatment
with recombinant CDK, showing that the observed inhibitory effects
are primarily dependent on the phosphorylation of MCM8-9 CDK
sites (Supplementary Fig. 3g-i). In contrast, the phosphorylation
status of MCM8-9 did not affect its affinity to DNA, as measured by

electrophoretic mobility shift assays (EMSA, Supplementary Fig. 3j).
In addition, phosphorylation of MCM8-9 did not alter its interaction
with HROB (Supplementary Fig. 3k). The regulation of the MCM8-9
helicase by phosphorylation resembled that of MCM4-6-7, which
was similarly found to be inhibited by CDK2-Cyclin A[44].

## MCM8-9 together with HROB unwinds branched DNA structures
We next turned to oligonucleotide-based substrates to better define
the MCM8-9 DNA unwinding preference. We observed that MCM8-9
and HROB unwound most efficiently a Holliday junction (HJ) followed
by a Y-structure, while 3' and 5' overhang, as well as fully double-
stranded DNA substrates were not unwound at all under our condi-
tions (Fig. 2a,b and Supplementary Fig. 4a). DNA unwinding of the HJ
substrate was marginally stimulated by human RPA, while RPA inhib-
ited the unwinding of the Y-structure, possibly because of competition
for the single-stranded region of the DNA substrate (Fig. 2a,b). DNA
unwinding was strongly stimulated by HROB in all cases (Fig. 2a). The
preferential unwinding of the Holliday junction substrate was unex-
pected, showing that in some cases a ssDNA tail is not required for the
MCM8-9 helicase. In summary, MCM8-9 and HROB clearly prefer to
unwind branched DNA structures.

The DNA unwinding specificity of the MCM8-9 and HROB
ensemble was at least in part a consequence of their DNA binding
preference: MCM8-9 bound with the highest affinity HJ and
Y-structured DNA (Fig. 2c and Supplementary Fig. 4b). Other sub-
strates, such as overhanged DNA, were bound only slightly less. HROB
also preferentially bound HJ and Y-structured DNA, while overhanged
or fully duplex DNA substrates were not bound almost at all (Fig. 2d
and Supplementary Fig. 4b).

Our observation that overhanged oligonucleotide-based DNA
structures were not unwound by MCM8-9, as opposed to a Y-junction,
suggests that the MCM8-9 complex requires the presence of a flap
structure at the junction point between ssDNA and dsDNA for DNA
unwinding. Similarly to our observation with MCM8-9, the outer *Sul-
folobus solfataricus* MCM surface was proposed to dynamically interact
with the 5'-tail of the strand being displaced, while the MCM ring
translocates in a 3'-5' direction on the opposite DNA strand[45], providing
a possible structural explanation for the DNA unwinding preference.
As our experiments with oligonucleotide-based DNA structures could
not be used to determine the DNA unwinding polarity of MCM8-9, we
turned to M13-based DNA paired with oligonucleotides forming
duplexes with short 5' or 3' tailed ssDNA flaps (Fig. 2e,f). We reasoned
that the very short flaps (5 nt) are unlikely to allow the loading of
MCM8-9; in fact, the truncated MCM8-9 complex failed to load on 10-
nt-long overhangs in a previous study[18]. We anticipated that MCM8-9
instead loads onto and translocates along the circular ssDNA, but the
direction of translocation was unknown. In case of 3'-5' translocation

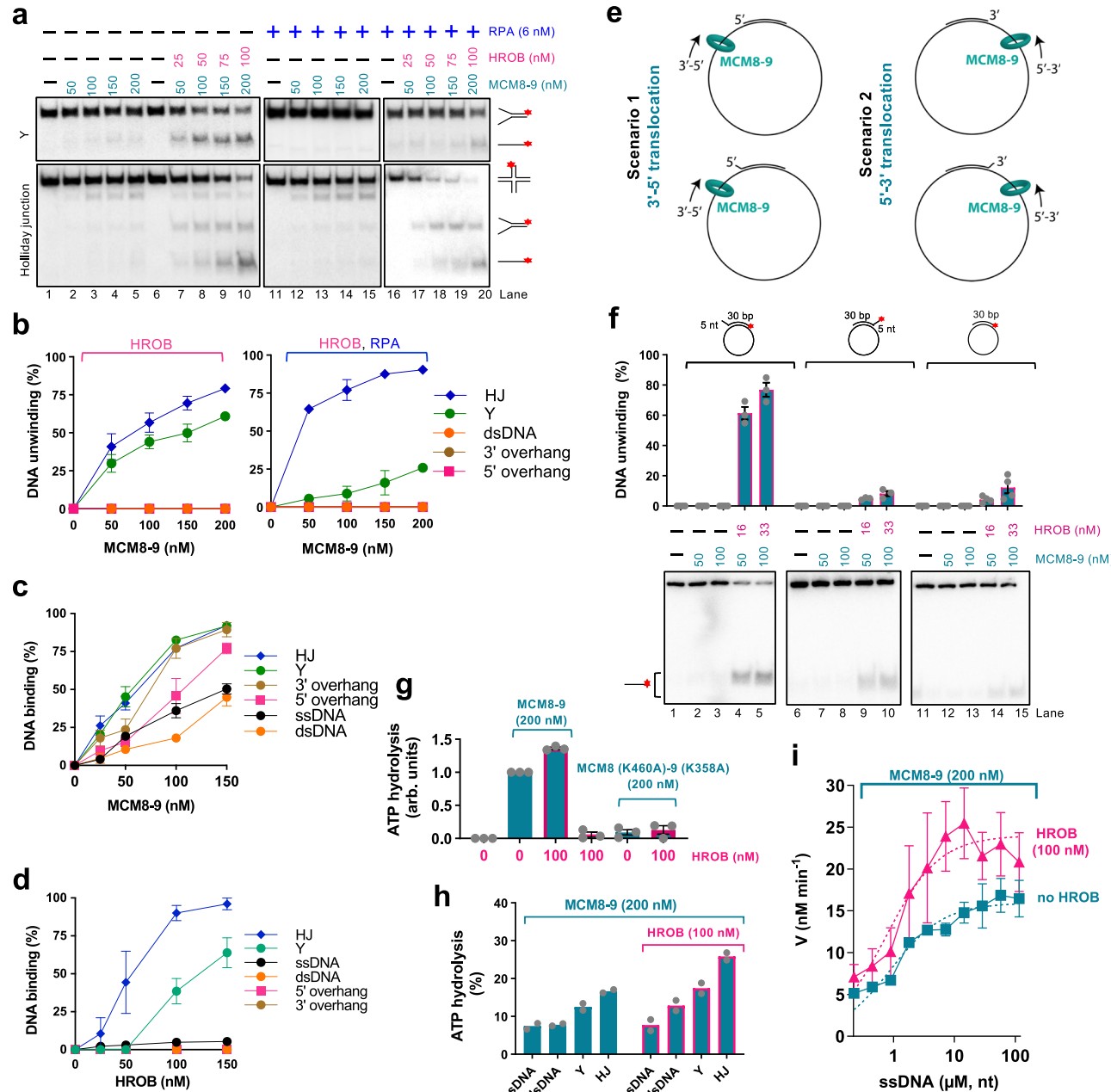

**Fig. 2 | MCM8-9 with HROB unwinds branched DNA structures. a** DNA unwinding by MCM8-9 without or with HROB and RPA, as indicated, with Y and Holliday junction (HJ) DNA substrates with 1 mM ATP, 5 mM magnesium acetate and 15 mM NaCl. Red asterisk, radioactive label. Representative of $n = 3$ independent assays is shown. **b** Quantification of helicase assays, such as shown in panel a and Supplementary Fig. 4a. Error bars, SEM; n = 3 independent experiments. **c** Quantification of DNA binding assays with MCM8-9, such as shown in Supplementary Fig. 4b. Error bars, SEM; $n = 3$ independent experiments. **d** Quantification of DNA binding assays with HROB such as shown in Supplementary Fig. 4b. Error bars, SEM; $n = 3$ independent experiments. **e** A cartoon representing two possible translocation polarities of MCM8-9 on M13-based substrates used in panel **f**. See text for details. **f** DNA unwinding by MCM8-9 with/without HROB, using M13-based substrates as indicated, and 30 mM NaCl. Top, quantification; error bars, SEM; bottom, representative of $n = 3$ independent experiments, except $n = 4$

independent experiments for no overhang substrate. Red asterisk, radioactive label. **g** Quantification of ATP hydrolysis (expressed as arbitrary units, arb. units, normalized to wild type MCM8-9 alone as 1) by 200 nM wild type MCM8-9 and ATP-binding deficient mutant MCM8 (K460A)-9 (K358A) with or without HROB. 7.2 μM (in nucleotides) M13-based circular ssDNA substrate was used as a co-factor. Error bars, SEM; $n = 3$ independent experiments. **h** ATP hydrolysis, expressed in % of total ATP hydrolysed by MCM8-9 with and without 100 nM HROB in the presence of various oligonucleotide-based DNA substrates (7.2 μM, in nucleotides) and RPA (0.58 μM). Bars show range; $n = 2$ independent experiments. **i** Relationship between ATP hydrolysis by MCM8-9 (200 nM) and the concentration of DNA (μM, in nucleotides) without and with HROB (100 nM). $V$ is the rate of ATP hydrolysis. Error bars, SEM; $n = 4$ independent experiments for [ssDNA] ranging from 0.225 to 3.6 μM, $n = 7$ for [ssDNA] ranging from 7.2 to 14.4 μM and $n = 5$ independent experiments for [ssDNA] ranging from 28.8 to 115.2 μM.

along the ssDNA, a short 5′ flap was expected to boost unwinding (scenario 1 in Fig. 2e), whereas in case of a 5′-3′ translocation, a 3′ flap would facilitate unwinding (scenario 2 in Fig. 2e). The experiments presented in Fig. 2f clearly support the scenario 1 and thus the 3′-5′

translocation polarity, in agreement with data obtained with truncated MCM8-9 without HROB previously[18]. The observed 3′-5′ translocation polarity of MCM8-9 is consistent with that of replicative MCM proteins from archaea or eukaryotes (MCM2-7)[46].

## HROB promotes the ATPase activity of MCM8-9

DNA unwinding by MCM8-9 and HROB depends on ATP hydrolysis[33]. We observed that MCM8-9 exhibited ATPase activity (Fig. 2g) that was strongly stimulated by DNA (Fig. 2h,i), which differs from replicative MCMs including MCM2-7 that display ATPase activity independently of DNA[10,12,47]. Mutation of the Walker A motif in both MCM8 (K460A) and MCM9 (K358A) subunits of the heterodimer drastically reduced ATPase activity, demonstrating that the observed activity is intrinsic to the MCM8-9 complex (Fig. 2g). MCM8-9-HROB showed the highest ATPase activity with Holliday junction and Y-structured DNA substrates, suggesting that the observed ATPase activity may reflect translocation of MCM8-9 along DNA, or DNA unwinding (Fig. 2h). We also note that ATPase activity was generally quite low.

HROB did not possess a detectable ATPase activity (Fig. 2g), as expected, and stimulated the ATPase of MCM8-9 $\sim$1.5-2-fold (Fig. 2h,i). The stimulatory effect of HROB on the MCM8-9 ATPase was notably less pronounced than its effect on DNA unwinding, suggesting that HROB makes the ATPase activity of MCM8-9 more productive. HROB was proposed to recruit MCM8-9 to DNA[31,33], and a modest stabilization of the protein-DNA species was indeed observed in the presence of HROB in electrophoretic mobility shift assays (Supplementary Fig. 3j). However, the EMSA assays are difficult to interpret as both MCM8-9 and HROB bind DNA, and it was not clearly apparent whether DNA binding was additive or synergistic (Supplementary Fig. 3j). To learn more about the effect of HROB on MCM8-9, we next varied the DNA concentration to obtain kinetic parameters for ATP hydrolysis by MCM8-9 without or with HROB (Fig. 2i). HROB stimulated the maximal rate of ATP hydrolysis ($V_{max}$) $\sim$1.5-fold, while the DNA concentration that supports 1/2 of $V_{max}$, corresponding to the affinity for DNA, was largely unaffected by HROB. These results suggested that HROB does not primarily recruit MCM8-9 to DNA, but may promote its activity post recruitment.

## Single-molecule experiments reveal low processivity of MCM8-9

To estimate the DNA unwinding processivity of MCM8-9 and HROB, we varied the length of the duplex DNA region annealed to M13-based ssDNA. The unwinding efficacy decreased $\sim$2-fold when the dsDNA region increased from 30 to 90 bp, suggesting that DNA unwinding by MCM8-9 and HROB is likely not very processive (Fig. 3a). In addition, we note that changing the ratio of MCM8-9 to HROB did not affect the processivity of MCM8-9 unwinding using the gel-based bulk assays (Supplementary Fig. 5a).

We next set out to define the helicase activity of the MCM8-9 and HROB ensemble more quantitatively using single-molecule magnetic tweezers. As opposed to ensemble gel-based techniques, the single-molecule approach allows to study individual active molecules, and it is unlikely to be biased by a proportion of potentially inactive proteins[48,49]. Considering that MCM8-9 efficiently unwinds HJs, we turned to a HJ construct with mobile arms of 262 bp in length (Fig. 3b). The unwinding or branch migration of the DNA arms would lead to a large DNA length difference, necessary to detect limited DNA unwinding (Fig. 3b and Supplementary Fig. 5b). Similarly to the ensemble measurements, we did not observe any activity of MCM8-9 or HROB individually, even at high concentrations (Supplementary Fig. 5c,d). However, the presence of both proteins in a 1:1 molar ratio resulted in short-ranged gradual unwinding events (Fig. 3c,d). No unwinding was observed in a buffer supplemented with non-hydrolysable ATP analog (ATP-γ-S), instead of ATP, demonstrating that the detected events were caused by active motor activity rather than DNA remodeling due to protein binding (Fig. 3e). Due to the symmetric nature of the Holliday junction, two distinct event types were detected. First, we observed gradual apparent DNA lengthening, with a mean velocity of $10 \pm 3$ bp/sec and a mean processivity of $41 \pm 5$ bp (Fig. 3b,c,f,g). We also observed gradual shortening events, with a mean velocity of $-(11 \pm 4)$ bp/sec and a mean processivity of

$-(40 \pm 7)$ bp (Supplementary Fig. 5b,e,f,g,). The first event types originate from the MCM8-9 complex translocating from the branching point in the direction of the Holliday junction arms (Fig. 3b), whereas the second type corresponds to the complex translocating in the direction of the Holliday junction stem (Supplementary Fig. 5b). We identified $\sim$75% of the events corresponding to the first event type (net upwards movement), whereas the remaining 25% belonged to the second type (net downwards movement). The non-symmetric occurrence of the two events most likely originates from the impact of the externally applied force (10 pN). Nevertheless, the mean unwinding velocity and processivity was similar in both cases (Fig. 3f,g and Supplementary Fig. 5f,g), showing that the applied force likely did not affect the motor function of the helicase ensemble. We note that the DNA unwinding velocities are comparable to reports for the yeast CMG helicase that translocates on ssDNA at 5–10 bp/sec[50] and that of $4.5 \pm 1.6$ bp/s for *Drosophila* CMG[51]. Also, in agreement with our bulk assays (Supplementary Fig. 5a), we note that changing the ratio of MCM8-9 to HROB in the single-molecule assays did not affect the processivity or the speed of DNA unwinding by MCM8-9 (Supplementary Fig. 6a-f). Our single-molecule experiments reinforce the notion that HROB stimulates the helicase function of the MCM8-9 complex and that MCM8-9-HROB per se is a helicase with a limited processivity.

## HROB promotes DNA unwinding by hexameric MCM8-9

MCM8 interacts with MCM9, and the stability of the proteins in human cell extracts is partially dependent on each other[17,28,35]. *Drosophila* only contains MCM8, raising questions whether complexes consisting of a single MCM8 or MCM9 human polypeptides may also be functional. We observed that MCM8 and MCM9 can be expressed and purified on their own from insect cells (Fig. 4a). Individually expressed MCM8 and MCM9 were largely monomeric as determined by mass photometry (Supplementary Fig. 7a,b). However, single MCM8 or MCM9, or a combination of the individually expressed proteins did not show any DNA unwinding activity, without or with HROB, in contrast to MCM8-9 proteins expressed and purified together as a complex (Fig. 4b,c). These experiments indicated that not only both MCM8 and MCM9 are required for DNA unwinding, but also that the co-expression of both subunits is necessary, likely to achieve proper folding (Fig. 4c). We also attempted to co-express HROB with MCM8-9, but this did not further increase the activity of the purified MCM8-9, suggesting that HROB is likely not required for the proper assembly of MCM8-9 upon expression (Supplementary Fig. 7c). As HROB interacts with MCM8-9 rather transiently, it could not be co-purified as a complex (Supplementary Fig. 7d).

We note that MCM8-9 and HROB unwound DNA in a concentration-dependent manner (Supplementary Fig. 7e). Interestingly, using a fixed concentration of MCM8-9, the highest stimulation of DNA unwinding was observed with a lower HROB concentration with respect to MCM8-9 (approximately 1:2-3 stoichiometric ratio) (Fig. 4d and Supplementary Fig. 7f). HROB did not promote DNA unwinding by the BLM helicase, but similarly became inhibitory together with BLM (Supplementary Fig. 7g) at higher concentrations. This experiment suggests that the inhibition of MCM8-9 by high HROB concentrations is non-specific, stemming likely from a competition for DNA.

We next tested the requirement of the ATPase activity of the individual MCM8 and MCM9 subunits for DNA unwinding. We observed that mutations of either the Walker A lysine residues K460A in MCM8 or K358A in MCM9 both reduced DNA unwinding of the heterocomplexes, although the relative contributions were different (Fig. 4e-g). No unwinding was observed with MCM8 (K460A)-9, while only moderately reduced DNA unwinding was observed with the MCM8-9 (K358A) variant, showing that the MCM8 ATPase is more critical. Nevertheless, the ATPase activities of both subunits are important for DNA unwinding (Fig. 4e-g).

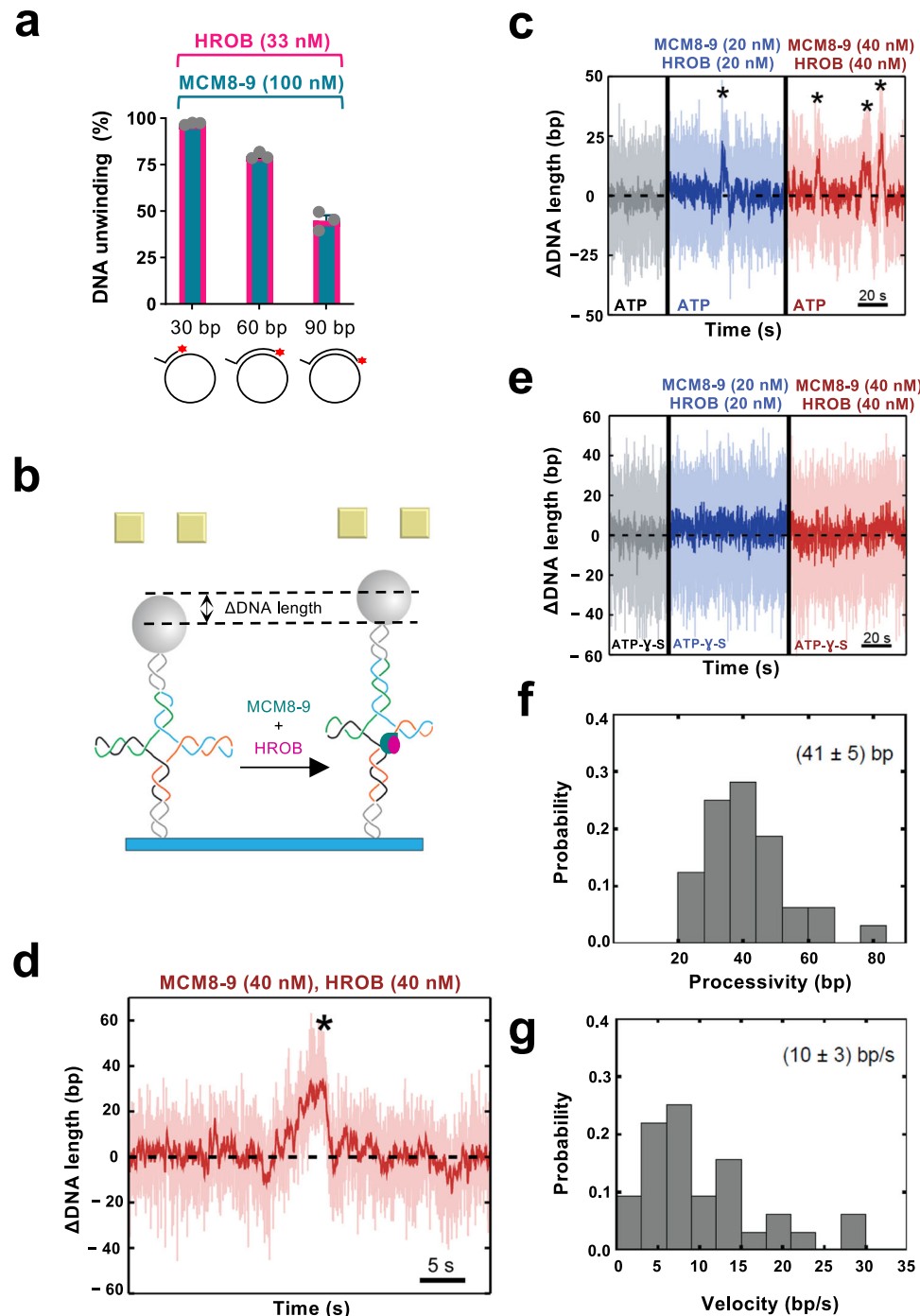

**Fig. 3 | Single-molecule analysis of the MCM8-9 helicase with HROB.**
**a** Quantitation of gel-based assays showing DNA unwinding by MCM8-9 (100 nM) and HROB (33 nM), using substrates with different lengths of duplex DNA (over-hang length 20 nt) as indicated, with 14 mM NaCl. Red asterisk, radioactive label. Error bars, SEM; $n = 3$ independent experiments. **b** Schematic representation of the magnetic tweezer setup and the investigated Holliday junction construct with 262 bp in each arm. When the protein ensemble is added, MCM8-9 with HROB can translocate in the direction of the arms, causing the measured DNA length to increase. **c** Activity of MCM8-9 and HROB with ATP, as indicated. Successive DNA unwinding events (highlighted by an asterisk), resulting in a net-increase of DNA length, were observed. **d** Magnified example trace for a representative unwinding event (highlighted by an asterisk) by MCM8-9 and HROB. **e** MCM8-9 with HROB do not unwind DNA with nonhydrolysable ATP-γ-S instead of ATP. **f.** Probability dis-tribution of DNA unwinding processivity by MCM8-9 with HROB, with a mean of $41 \pm 5$ bp, of events leading to DNA extension. Error, 2SEM; DNA unwinding of 32 molecules was measured. **g** Probability distribution of DNA unwinding velocity by MCM8-9 with HROB, with a mean of $10 \pm 3$ bp sec$^{-1}$, of events leading to DNA extension. Error, 2SEM; DNA unwinding of 32 molecules was measured.

ATPase sites of AAA+ helicase family members are formed at the interface of two polypeptides, and a hexamer formation is thought to be necessary for DNA unwinding[10–12]. Our observation that the integrity of both ATPase sites within the complex is important suggests that both the MCM8-MCM9 and MCM9-MCM8 interfaces are required for productive DNA helicase activities (Fig. 4e), although to different extents. Therefore, the MCM8-9 species active in DNA unwinding is of a higher order than a dimer, in agreement with AlphaFold2 modeling (Fig. 1i, Fig. 4e and below). Mass photometry measurements of our recombinant MCM8-9 complex showed a mixed population of dimers,

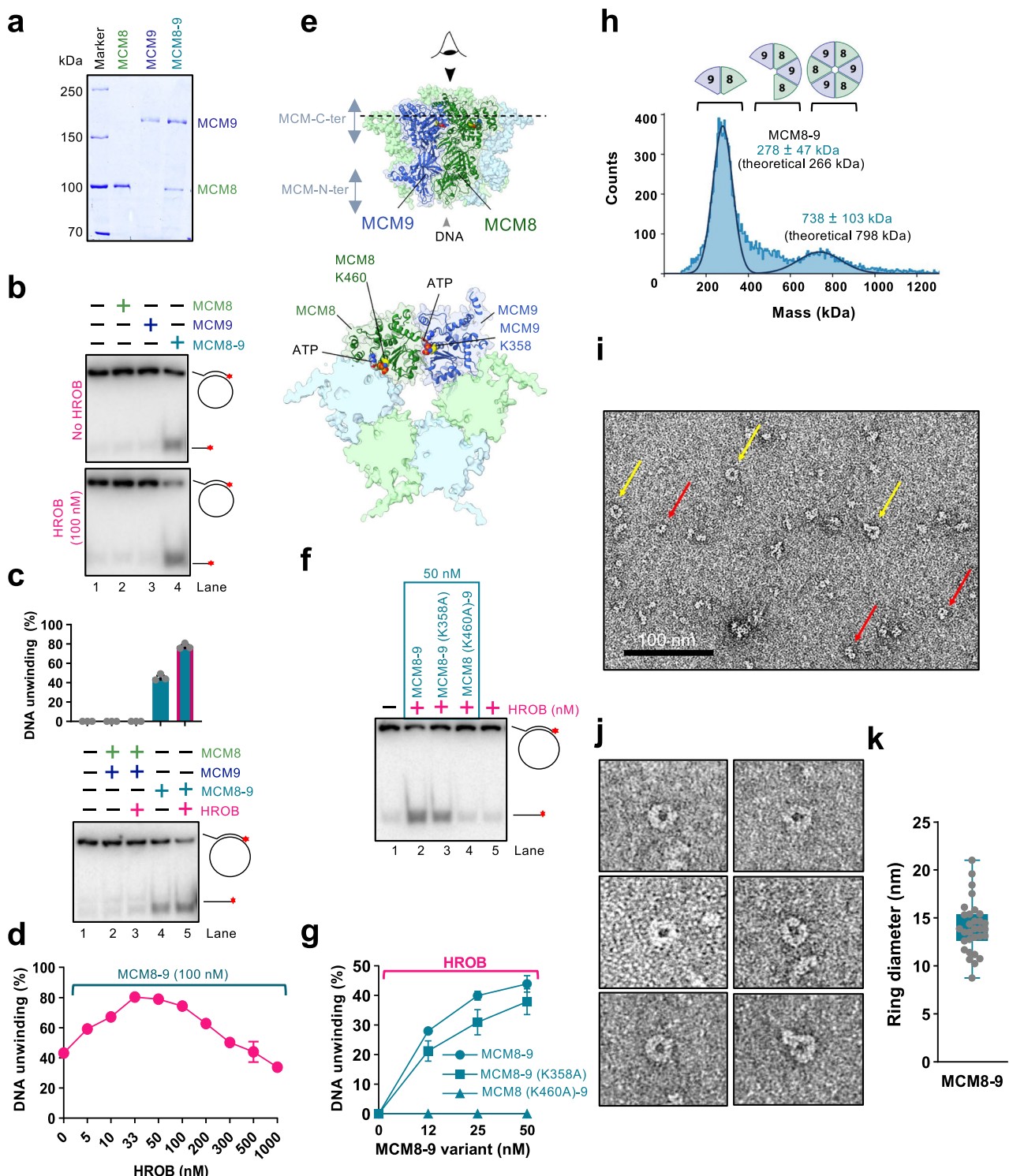

tetramers, and hexamers (Fig. 4h). We note that the oligomeric state of MCM8-9 in solution did not change upon incubation with ATP-γ-S (Supplementary Fig. 7h,i). We next performed imaging of the complexes by negative staining transmission electron microscopy (Fig. 4i). Although we observed a variety of sizes and shapes, in accord with the mass photometer measurements, a small fraction of the protein complexes exhibited a ring-like structure with a cavity in the center (Fig. 4j,k), in agreement with the AlphaFold2 model (Fig. 4e). The mean diameter of the clearly visible MCM8-9 rings in the negative staining transmission electron microscopy images was 14 nm, which is similar to the mean ring diameter measured in cryoEM images of full-length

chicken MCM8-9 hexamer (13.2 nm)[20], truncated human MCM8-9 (14 nm)[18] or the MCM2-7 hexamer from budding yeast (13.3 nm)[52].

**ATP locks the MCM8-9 ring on ssDNA, irrespectively of HROB**
To study the effects of MCM8-9 oligomerization on its biochemical activities, we next performed size exclusion chromatography. The elution profile showed a rather wide distribution without clearly defined peaks, likely reflecting the dynamic nature of the MCM8-9 complexes. Nevertheless, a pool of early eluates showed a higher proportion of hexamers (hexamer-rich sample, ~1:1 dimer:hexamer in protein counts or 1:3 in protein mass) (Fig. 5a,b), as opposed to a

**Fig. 4 | HROB promotes DNA unwinding by hexameric MCM8-9.**
**a** Representative gel showing purified FLAG-tagged MCM8, MBP-tagged MCM9 and MCM8-9, $n = 2$ independent experiments. **b** MCM8, MCM9 (single polypeptides, 100 nM) and MCM8-9 (co-expressed, 100 nM) were used in helicase assays with/without HROB (100 nM) with M13-based circular DNA at 21 mM NaCl. Red asterisk, radioactive label. Representative of $n = 3$ independent experiments is shown. **c** MCM8 and MCM9, combined upon individual expression (lanes 2 and 3) or co-expressed (lanes 4 and 5), all 100 nM, were used in helicase assays with or without HROB (100 nM) using M13-based circular DNA at 21 mM NaCl. Red asterisk, radio-active label. Top, quantification; error bars, SEM; bottom, representative of $n = 3$ independent experiments. **d** Quantitation of DNA unwinding by MCM8-9 in the presence of HROB such as shown in Supplementary Fig. 7f. Error bars, SEM; $n = 3$ independent experiments, except for [HROB] = 0, 200 and 300 nM, $n = 6$. **e** AlphaFold2 model depicting ATPase sites at the interfaces of MCM8 and MCM9 within the hexamer. Walker A lysines of MCM8 (K460) and MCM9 (K358) are

shown. The lower cartoon represents a C-terminal end view. **f.** DNA unwinding by wild type or ATP binding-deficient MCM8-9 variants (all 50 nM) with HROB (17 nM), using M13-based circular DNA with 25 mM NaCl. Red asterisk, radioactive label. Representative of $n = 3$ independent experiments is shown. **g** Quantification of helicase assays such as shown in panel f. HROB was used at 1/3 concentration of MCM8-9. Error bars, SEM; $n = 3$ independent experiments. **h** Measured molecular weight distribution of MCM8-9 (FLAG-tagged MCM8 and MBP-tagged MCM9) displaying dimers and hexamers by mass photometry. Error, SD. **i** Negative staining transmission electron micrograph of MCM8-9 (220 nM). Yellow arrows indicate clearly distinguishable top views of MCM8-9 rings, while red arrows denote smaller, less well-recognizable rings or side views of rings. Representative of $n = 3$ independent images is shown. **j** Representative transmission electron micrographs of MCM8-9 rings. **k** Quantification of diameter sizes of distinguishable MCM8-9 rings as shown in panel j. Error bars, range; 31 MCM8-9 rings were measured.

---

dimer-rich sample (~3:1 dimer:hexamer in protein counts and 1:1 in protein mass) (Figs. 5a and 4h). Contrary to our expectations, the dimer-rich sample showed a higher apparent DNA unwinding activity than the hexamer-rich preparation together with HROB. Furthermore, the hexameric complex was unstable and prone to aggregation showing that the MCM8-9 hexamers that assembled in solution without the DNA substrate may be inactive complexes (Fig. 5c and Supplementary Fig. 8a).

We next used the more active MCM8-9 dimer-rich preparation and analyzed its binding to circular and linear ssDNA, without or with ATP and without or with HROB. Several important conclusions can be made from these electrophoretic mobility shift experiments. First, MCM8-9 showed enhanced binding to circular ssDNA in the presence of ATP, as opposed to reactions without ATP (Fig. 5d,e). The enhanced binding in the presence of ATP was not observed when we used the ATP binding-deficient MCM8 (K460A)-9 (K358A) mutant complex (Fig. 5f), demonstrating that the effects result from direct ATP binding to the ATPase sites of MCM8-9. Similar effects were noted when using ATP-γ-S instead of ATP with the wild type protein complex, showing that ATP binding, and not ATP hydrolysis, stabilizes the MCM8-9 complex on DNA (Supplementary Fig. 8b). These data mirror the behavior of MCM2-7, which was biochemically demonstrated to form rings on DNA[12,53]. Next, the stabilization of MCM8-9 on DNA in the presence of ATP was not observed on linear ssDNA (Supplementary Fig. 8c) or on circular dsDNA (Fig. 5g). While a ring can slide off the ends of linear DNA, it remains topologically locked in place on circular DNA. These results thus suggest that ATP helps close the MCM8-9 ring around ssDNA. Finally, HROB did not affect the capacity of the MCM8-9 ring to lock onto ssDNA (Fig. 5h). We conclude that in the presence of ATP, MCM8-9 forms rings on ssDNA independently of HROB, and that the primary function of HROB is therefore downstream of MCM8-9 loading and locking onto ssDNA. Oligomerization and circularization of MCM8-9 in solution without DNA may yield inactive protein complexes.

## MCM8-9 rings assemble from heterodimers on ssDNA and form two interfaces

As described above, the MCM8-9 hexamer predicted by the Alpha-Fold2 model contains two distinct interfaces repeating alternatively three times along the ring structure (Fig. 6a,b, upper part). We denote the first one as interface I. When viewed from the C-terminal side of the toroid, interface I lies between subunits MCM8 and MCM9 (dashed purple line in Fig. 6a). The second, interface II, is shown by the dashed orange line in Fig. 6b between MCM9 and MCM8. HROB binds MCM8 and MCM9 across interface I (Fig. 1i). When modeling the dimer between MCM8 and MCM9, AlphaFold2 only generated interface I, suggesting that the evolutionary information encoding its structure is much stronger than that of the interface II. In each of these two interfaces, there are multiple contact points between the adjacent

subunits at the N-terminal, central, and C-terminal parts of the proteins. Several mutants were designed to probe the relative importance of interfaces I and II to assess the complex formation and effects on DNA unwinding (Fig. 6a,b, lower part).

Interface I was mutated at five well-exposed positions in the structure of each monomer (Fig. 6a). First, two mutations caused charge swapping of the salt-bridge residues at the interface between the N-terminal domains, MCM8 (E314R)-9 (R256E); second, we mutated one highly conserved position in the pre-sensor-1-β-hairpin of MCM8 contacting MCM9, creating MCM8 (V547Q)-9[54] and third, two positions in the parallel helices mediating the interaction between the C-terminal domains were replaced, creating MCM8 (R697D)-9 (L506E). None of the single or double mutants produced in the interface I could be purified as a heterocomplex, suggesting that the destabilization of this interface is highly detrimental to the formation of a stable dimer.

In contrast, a set of five positions mutated similarly at interface II (Fig. 6b) had no detrimental effect on the production of the MCM8-9 heterodimer, and mutant proteins could be purified for biochemical analysis. The mutation located in the central pre-sensor-1-β-hairpin of MCM9, MCM9 (V444Q), structurally equivalent to MCM8 (V547Q), only partially reduced the DNA unwinding activity, suggesting that this position in MCM9 is less constrained than its counterpart in MCM8 (Supplementary Fig. 9a). Interface II mutants in the C-terminal domains, MCM8 (I610E)-9 (L547E) were defective in DNA unwinding (Supplementary Fig. 9a). The C-terminal domains of MCM proteins harbor the ATPase/helicase domains, which likely explains the detrimental effect of these mutations.

The most interesting were the mutants in interface II disrupting a salt-bridge between the N-terminal domains of MCM8 and MCM9, MCM9 (D230R) and MCM8 (R309D) (Fig. 6b). We observed that individually or combined, the mutations strongly reduced hexamer formation in solution, as apparent from mass photometry experiments (Fig. 6c), suggesting that the more labile interface II mediates the assembly of dimers into hexamers. HROB bound to the wild type MCM8-9 complex similarly as to the MCM8 (R309D)-9 mutant, while the binding to the MCM8 and MCM9 subunits alone was much weaker (Supplementary Fig. 9b). These data reinforce the notion that HROB binds across the stable interface I (Figs. 1i and 6a), which is not disrupted by the interface II mutation R309D in MCM8 (R309D)-9.

Interestingly, while the combined MCM8 (R309D)-9 (D230R) heterodimer was as active as the wild type complex in DNA unwinding (Fig. 6d and Supplementary Fig. 9a), the individual single point mutants, MCM8 (R309D)-9 and MCM8-9 (D230R) showed notably elevated activity in conjunction with HROB, compared to wild type MCM8-9 (Fig. 6d). These data suggest that a moderate destabilization of the N-terminal part of interface II creates a more dynamic protein complex, which prevents the formation of inactive hexamers without DNA, at least in vitro. The MCM8 (R309D)-9 or MCM8-9 (D230R) mutants are then more likely to assemble into a productive helicase on

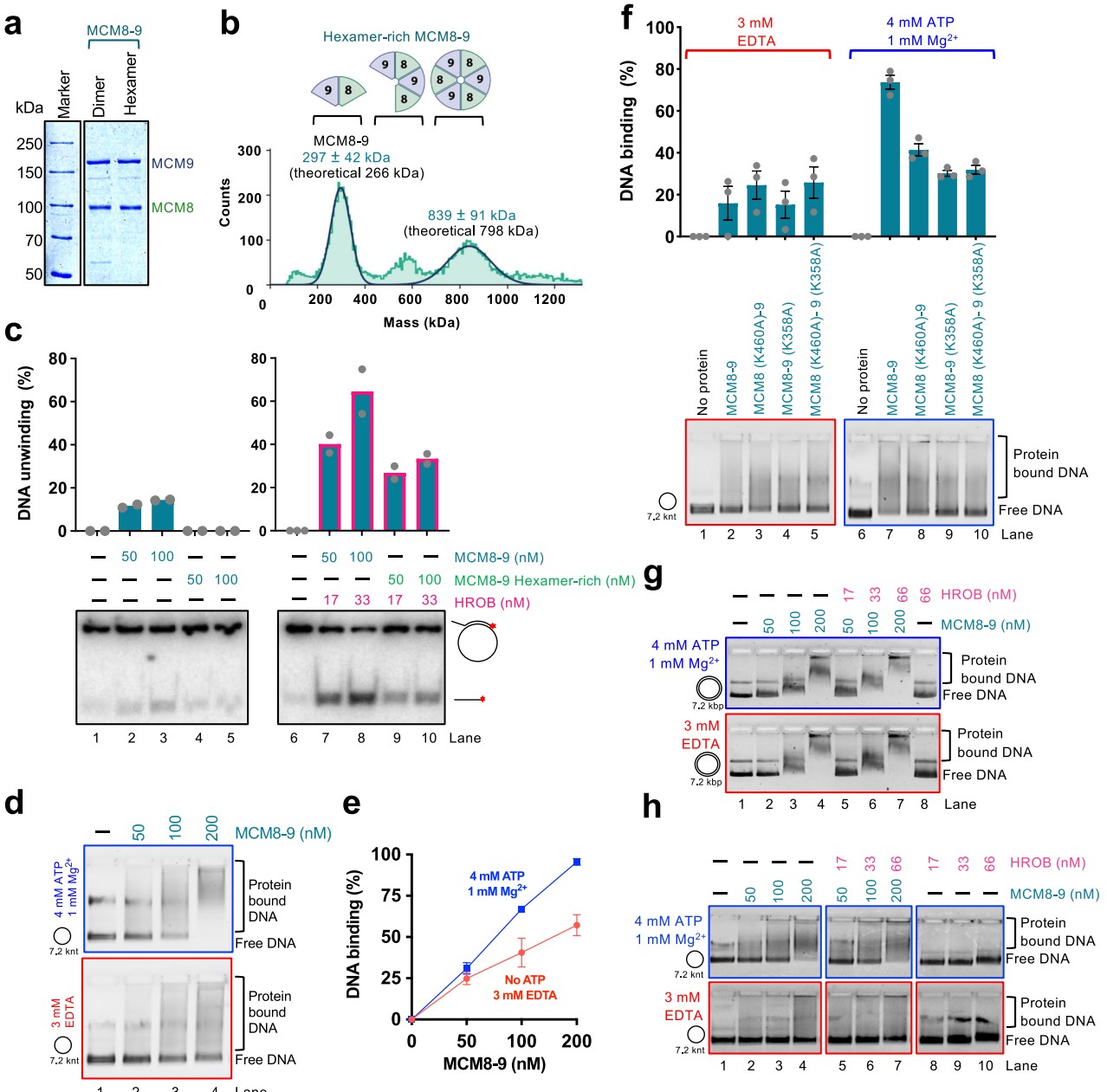

**Fig. 5 | ATP locks the MCM8-9 ring on ssDNA, irrespectively of HROB.**
**a** Representative gel showing purified wild type standard (dimer-rich, dimer) and hexamer-rich (hexamer) MCM8-9 preparation, $n = 2$ independent experiments. **b** Measured molecular weight distributions of hexamer-rich MCM8-9, purified by size exclusion chromatography by mass photometry. MCM8 was FLAG-tagged and MCM9 was MBP-tagged. Compare with Fig. 4h. Error, SD. **c** DNA unwinding using standard, dimer-rich (Fig. 4h) and hexamer-rich (Fig. 5b) preparations of MCM8-9, with and without HROB, as indicated, using M13-based circular ssDNA substrate with 14 mM NaCl. Red asterisk, radioactive label. Top, quantification; bars show range; bottom, representative of $n = 2$ independent experiments. **d** Electrophoretic mobility shift assays with human MCM8-9, with or without ATP, as indicated, using

M13-based circular ssDNA substrate. Representative of $n = 3$ independent experiments is shown. **e** Quantification of assays such as shown in panel d. Error bars, SEM; $n = 3$ independent experiments. **f** Electrophoretic mobility shift assays with ATP-binding deficient variants of human MCM8-9 (100 nM), with or without ATP, as indicated, using circular M13-based ssDNA. Top, quantification; error bars, SEM; bottom, representative of $n = 3$ independent experiments. **g** Electrophoretic mobility shift assays with MCM8-9, with and without ATP, as indicated, using M13-based circular dsDNA as a substrate. Representative of $n = 3$ independent experiments is shown. **h** Electrophoretic mobility shift assays with human MCM8-9, with and without HROB, with and without ATP, as indicated, using circular M13-based ssDNA as a substrate. Representative of $n = 3$ independent experiments is shown.

DNA, leading to overall elevated activity. Considering that the more active species MCM8 (R309D)-9 and MCM8-9 (D230R) were hyperactive in helicase assays yet dimeric in solution, the data suggest that an active hexameric helicase assembles on DNA from the heterodimeric building blocks. In accord with these conclusions, we observed that the dimeric MCM8 (R309D)-9 mutant formed higher-order oligomers, including those corresponding to the molecular weight of the hexamer, only in the presence of both DNA and ATP/Mg²⁺

(Supplementary Fig. 9c-f). These results reinforce the notion that MCM8-9 unwinds DNA as a hexamer, which is further supported by our AlphaFold2 modeling and recent structural studies[18,20].

## Labile interface II between MCM9 and MCM8 subunits powers DNA unwinding
As introduced above, an active ATPase site of the MCM helicases is reconstituted from residues belonging to the interface of neighbor

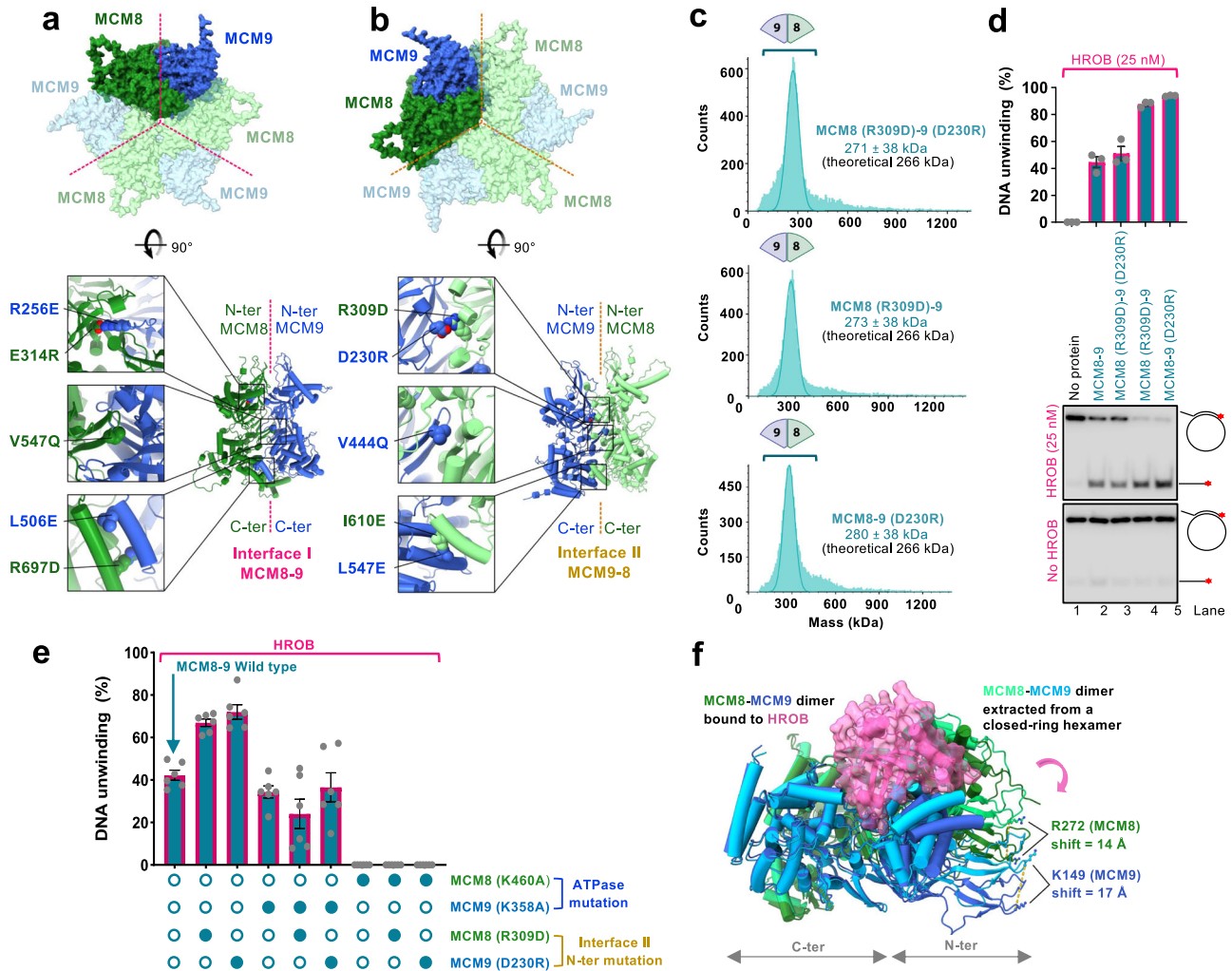

**Fig. 6 | MCM8-9 rings assemble from heterodimers via two distinct interfaces.**
**a** AlphaFold2 model showing a C-terminal view of the MCM8-9 hexamer (top). Interface I is indicated with a purple dotted line. Bottom part, mutations disrupting interface I were designed in the N-terminal, central, and C-terminal part of the interface. **b** AlphaFold2 model showing a C-terminal view of the MCM8-9 hexamer (top). Interface II is indicated with an orange dotted line. Bottom part, mutations disrupting interface II were designed in the N-terminal, central, and C-terminal part of the interface. **c** Molecular weight distributions of human MCM8-9 variants with disrupted N-terminal part of interface II measured. Compare with wild type MCM8-9 in Fig. 4h. MCM8 was FLAG-tagged and MCM9 was MBP-tagged. **d** DNA unwinding by human MCM8-9 variants with disrupted N-terminal part of interface II (50 nM) and HROB (25 nM) using circular M13-based DNA at 25 mM NaCl. Red asterisk,
radioactive label. Top, quantification; error bars, SEM; bottom, representative of n = 3 independent experiments. **e** DNA unwinding by human MCM8-9 variants (25 nM) with disrupted N-terminal part of interface II, in combination with ATPase Walker A mutations, as indicated, in the presence of HROB (8 nM) at 25 mM NaCl. The MCM8-9 variants were purified as complexes of MCM8 and MCM9 containing the mutations as indicated by the dots at the bottom of the panel. Full dots indicate the presence of corresponding mutations in the complex and empty dots show absence of mutations (i.e. wild type). The panel shows quantification of assays such as shown in Supplementary Fig. 10a-c; error bars, SEM; n = 6 independent experiments. **f** AlphaFold2 model depicting a structural change of the MCM8-9 dimer upon binding of HROB. For simplicity, only MCM8-9 dimer is shown.

subunits[10–12]. The catalytic lysine of the Walker A motif from the MCM8 subunit K460 is predicted to lie at the labile interface II between MCM9 and MCM8, while the K358 of MCM9 maps to the stable interface I between MCM8 and MCM9 (Figs. 4e and 6a,b). As the integrity of K460 in MCM8 is more important for the helicase activity of the complex (Fig. 4f,g), the data show that ATP hydrolysis at the labile interface II is more important for DNA unwinding. Therefore, although the majority of purified MCM8-9 forms stable dimers in solution mediated by the stable interface I, helicase activity is dependent on the interface II that is only assembled within the higher-order complex.

To confirm the relative importance of the ATPase sites within the two interfaces, we next combined the mutations destabilizing the N-terminal part of interface II (R309D in MCM8, or D230R in MCM9) with the Walker A mutants in the C-terminal domains of MCM8 (K460A, affecting interface II) and in MCM9 (K358A, affecting interface

I), within the same heterodimers (Fig. 6e). Disruption of the ATPase site in the labile interface II completely disrupted unwinding activities, while disruption of the ATPase site in the stable interface I had only a moderate effect, in all mutant combinations tested (Fig. 6e and Supplementary Fig. 10a-f). Collectively, these data support our model that interface I mediates stable interaction between MCM8 and MCM9, and it thus has a more structural function. Instead, the more dynamic interface II is involved in the assembly of the heterodimers into active hexameric complexes on DNA. The ATPase site at this dynamic interface contributes disproportionally more to the unwinding activity of the MCM8-9 ensemble.

## Discussion
We employed a combination of structure prediction, mutagenesis and ensemble as well as single-molecule biochemical approaches to gain insights into the mechanism of DNA unwinding by MCM8-9 and its

regulation by HROB. We demonstrate that MCM8 and MCM9 proteins in solution form diverse oligomeric species ranging from dimers to hexamers, in agreement with recent structural studies focused on chicken MCM8-9[20] or truncated human MCM8-9 complex[18]. Our biochemical data suggest that MCM8-9 hexamers are necessary for DNA unwinding in conjunction with HROB. Hexamers assembled in solution without DNA are not active, and we cannot exclude that these species form artificially during protein overexpression. We provide biochemical evidence that MCM8-9 assembles on ssDNA into an active hexameric helicase from heterodimeric building blocks in the presence of ATP. HROB dramatically stimulates all DNA unwinding activities of MCM8-9, which depends on extensive contacts with both MCM8 and MCM9 subunits of the ensemble. Interestingly, the assembly and DNA loading of MCM8-9 is not stimulated by HROB; rather, HROB preferentially stimulates DNA unwinding downstream of MCM8-9 loading and ring formation.

The ATPase sites of all MCM hexameric helicases form at the protein interfaces from residues belonging to adjacent subunits[10–12]. The MCM8-9 ring involves two distinct protein-protein interfaces, each repeating three times between the alternating MCM8 and MCM9 subunits along the ring. We find that one of these interfaces (interface I) is stable, and mediates a constitutive heterodimer formation. HROB is predicted to bind across this stable interface to the MCM8-9 heterodimer. We could detect stochiometric binding of HROB to MCM8-9 heterodimer, but we could not obtain experimental evidence for how many HROB molecules can simultaneously bind to a MCM8-9 hexamer. Mutations within the stable interface destabilize the dimer so that it cannot be purified. Interestingly, disruption of the ATPase site located within interface I has only a moderate effect on the DNA unwinding activity, suggesting that interface I has primarily a structural function. Interface II instead mediates the assembly of the heterodimers into hexamers. Disruption of this second interface allows the preparation of the heterodimer and instead reduces the fraction of non-productive hexamers detected in solution. The ATPase site located at this second labile interface is instead essential for DNA unwinding.

Our hypothesis that an active hexameric helicase assembles from dimers on DNA is supported by several lines of biochemical evidence. First, size exclusion chromatography-based enrichment of hexameric MCM8-9 complexes formed in solution without DNA revealed that they are largely inactive (Fig. 5a-c). ATP helps to lock the MCM8-9 rings on ssDNA (Fig. 5d,e), showing that DNA is necessary for productive complex formation. Second, we find that mutations within the N-terminal region of the labile interface II (such as R309D in the MCM8 subunit) reduce the fraction of non-productive hexamers in solution but stimulate overall DNA unwinding. Mass photometry experiments showed that while the MCM8 (R309D)-9 mutant is almost exclusively dimeric per se, it forms hexameric complexes in the presence of DNA and ATP/Mg$^{2+}$. Third, ATP hydrolysis mediated by interface II is absolutely essential for DNA unwinding, as opposed to the ATPase site formed between the subunits of a stable dimer (Figs. 4e, f, 6a, b). Therefore, species of higher order than dimers are essential for DNA unwinding. The conclusions based on our biochemical analysis are in agreement with the AlphaFold2 models presented here. While this manuscript was in revision, two studies published cryoEM structures of MCM8-9 hexameric complexes. One study reported the structure of human ADP-bound MCM8-9 (MCM9 was lacking the C-terminal extension, PDB: 8S91)[18], while the other paper reported the structures of chicken MCM8-9 complex, and human MCM8-9 N-terminal domain ring[20]. Our AlphaFold2 model closely resembles particularly the experimental 8S91 structure[18], especially when comparing the structures of N-terminal and C-terminal rings independently (Supplementary Fig. 11a). The low root mean square deviation (RMSD) values for the N- and C-terminal rings (1.9 vs. 2.2 Å) demonstrated the general reliability of the modelled assembly.

Although our biochemical experiments and available CryoEM structures[18,20] point towards MCM8-9 operating as a hexamer, the data are not entirely unambiguous that hexamer is the active form under all conditions.

HROB was initially proposed to recruit MCM8-9 to DNA, as HROB depletion disrupted the accumulation of MCM8 but not vice versa[31,33]. HROB binds branched DNA, similarly to MCM8-9, and a weak enhancement of protein binding when both HROB and MCM8-9 were combined was observed, particularly under more restrictive conditions (Supplementary Fig. 3e). It is possible that the recruitment function is more important in cells, where MCM8-9 needs to compete with other cellular factors. Nevertheless, our data clearly demonstrate a function of HROB that is independent of MCM8-9 recruitment. We found that ATP promotes the loading and locking of the MCM8-9 ring on ssDNA, a process that is unaffected by HROB (Fig. 5d-h). This observation is in agreement with the AlphaFold2 modeling coupled with mutagenesis that position HROB across the stable interface I through functionally important contacts with both MCM8 and MCM9 subunits (Fig. 1i-k), which provides a structural explanation for why HROB does not stabilize MCM8-9 ring on DNA (Fig. 5h).

How can HROB stimulate DNA unwinding by MCM8-9 downstream of its assembly on DNA? Our biochemical data do not provide direct evidence to answer this question, but the AlphaFold2 model together with the recent cryoEM study of the chicken complex[20] allow us to propose a hypothetical model. We noted that AlphaFold2 model predicts a rotation of both MCM9 and MCM8 N-terminal subunits with respect to their C-terminus upon HROB binding (Fig. 6f). Considering two highly conserved basic residues in the N-terminal domains well positioned to bind DNA at the entrance of the helicase core, MCM9 (K149) and MCM8 (R272), their position is predicted by AlphaFold2 to shift by 17 Å and 14 Å, respectively (Fig. 6f). Accordingly, when observing the relative orientations of the N-terminal and C-terminal rings, we have observed larger variations of our AlphaFold2 model with respect to the experimental structure of human ADP-bound MCM8-9 (Supplementary Fig. 11b, PDB: 8S91)[18], likely reflecting the flexibility and conformational states during the functional cycles of the enzyme. We speculate that repetitive structural changes in the relative orientation of the MCM8-9 N- and C-terminal rings induced by HROB may activate the helicase ensemble. We also observed that ATP hydrolysis by MCM8-9 is comparatively weakly stimulated by HROB (Fig. 2g-i), in contrast to DNA unwinding (Fig. 2f), where the stimulatory effect of HROB is much greater. Therefore, HROB facilitates productive ATP hydrolysis, possibly by coordinating the conformational changes with DNA translocation and unwinding. Additionally, a positively charged patch on HROB near the MCM8 subunit may mediate interaction with the displaced DNA strand. A surface patch of archaeal MCM was similarly proposed to explain the binding and unwinding preference for Y-shaped DNA[45]. The activation of MCM8-9 by HROB downstream of its loading may thus be reminiscent of MCM2-7 activation by Ccd45-GINS within the replicative CMG helicase complex[5,7,8], or by MCM10[55,56].

## Methods

### Cloning, expression, and purification of recombinant proteins
Human MCM8-9 and HROB were expressed in *Spodoptera frugiperda* 9 (*Sf*9) insect cells[33], using pFB-MBP-MCM9 or pFB-FLAG-MCM8, as well as pFB-MCM8IP-FLAG, respectively. Please note that MCM8IP is the previous name for HROB[33]. The protein sequence of HROB-FLAG is shown in Supplementary Table S1. The used HROB sequence is the closest to isoform 4 (Q8N3J3-4, UniProt), because it lacks Q-484. However, the sequence also contains a common polymorphism, T-126 → P-126. We note that the isoform 1 sequence of HROB (UniProt, Q8N3J3-1) also stimulates DNA unwinding by MCM8-9, albeit to a lower extent. The AlphaFold2 modeling was done using isoform 1 and therefore, all amino acid residue numbers used for HROB in the

manuscript refer to their positions in the sequence of isoform 1. The MCM8-9 proteins were purified using MBP- and FLAG-tag-based affinity chromatography as previously done, except that the MBP-tag was not cleaved[33]. Unless indicated otherwise, MCM8 and MCM9 were co-expressed and co-purified as a complex. Mutations in the expression vectors were generated using the QuikChange II XL site-directed mutagenesis kit, following manufacturer's instructions (Agilent Technology). Sequences of all primers used for site-directed mutagenesis are listed in Supplementary Table S2. The MCM8-9 mutants were expressed and purified as the wild type complex. Wild type HROB and mutants were purified using FLAG M2 Affinity Gel (Sigma, A2220) based column chromatography[33].

For the expression of phosphorylated MCM8-9 (pMCM8-9) variants, *Sf*9 cells were treated with 50 nM Okadaic acid (APExBIO, A4540) to preserve proteins in their phosphorylated state for 3 h before harvesting. Additionally, the cell extracts were supplemented with 50 nM Okadaic acid (APExBIO), 1 mM sodium orthovanadate (Sigma), 20 mM sodium fluoride (Sigma), and 15 mM sodium pyrophosphate (Applichem) during lysis to preserve the phosphorylation status. Where indicated, pMCM8-9 was dephosphorylated with λ-phosphatase (New England Biolabs) according to the manufacturer's instructions. For "mock" controls, λ-phosphatase was excluded from the reactions, and the sample was otherwise incubated as the λ-phosphatase-treated reactions. Similarly, λ-MCM8-9 was phosphorylated by CDK2-CycA, where indicated, using standard conditions of in vitro phosphorylation[57]. The hexamer-rich prep of MCM8-9 was purified using the standard method for MCM8-9 purification, except by adding a size exclusion chromatography step (Sephacryl S-400 HR, GE Healthcare) in between the amylose and FLAG steps, and pooling the early eluting MCM8-9 fractions. Human RPA was expressed in *E. coli* and purified using ÄKTA pure (GE Healthcare) with HiTrap Blue HP, HiTrap desalting and HiTrap Q chromatography columns (all GE Healthcare)[58].

### DNA substrate preparation
For DNA binding experiments with linear single-stranded DNA, oligonucleotide 314 (25 nt), PC1253 (50 nt), X12-3 HJ3 (93 nt) was labeled at the 3′ terminus with [α-$^{32}$P] dCTP (Hartmann-Analytic) by terminal transferase (New England Biolabs), according to standard protocols[59]. The oligonucleotide-based DNA substrates used for DNA binding, ATPase and DNA unwinding experiments were ssDNA (labeled oligonucleotide PC1253), dsDNA (labeled PC1253 and PC1253C), 5′ overhang (labeled PC1253 and 312), 3′ overhang (labeled PC1255 and 314), Y-structure (labeled PC1254 and PC1253) and Holliday junction (labeled PC1253, PC1254, PC1255 and PC1256)[60]. The indicated oligonucleotides were labeled at the 3′ terminus with [α-$^{32}$P] dCTP (Hartmann-Analytic) by terminal transferase (New England Biolabs) prior to annealing, according to standard procedures. Unincorporated nucleotides were removed using Micro BioSpin™ P-30 Gel Columns (Bio-Rad). Plasmid length DNA binding experiments were performed with unlabeled circular or linearized M13mp18 single (New England Biolabs, M13mp18 single-stranded DNA) or double-stranded DNA (New England Biolabs, M13mp18 RF I DNA). PC216[61] was used for mass photometry-based experiments, where indicated.

A major portion of the DNA unwinding experiments (unless indicated otherwise) were performed using 5′ overhang-M13-based circular DNA. This was prepared using oligonucleotide (M13-5′-dT 40 overhang) containing a 40 nt-long tail at the 5′ end, as well as a 37 nt-long region complementary to the M13mp18(+) strand (nucleotides 6289−6326), which was annealed to M13mp18 ssDNA[33]. Variants of the annealed oligonucleotide with either no overhang (M13-37mer no overhang) or a 40 nt-long tail at the 3′ end (M13-3′-dT 40 overhang) were used to prepare the no overhang or 3′ overhanged-M13 based circular DNA substrates, respectively. Similarly, other variants were prepared by altering the length of the annealing region to produce

substrates with 30, 60 and 90 bp-long complementary regions to the M13mp18(+) strand (oligonucleotides M13-5′-dT20overhang-30bp_c, M13-5′-dT20overhang-60bp_c and M13-5′-dT20overhang-90bp_c) with 20 nt-long tail at the 5′ end. Another set of variants was prepared by altering both the length of the annealing region and the non-complementary tail (oligonucleotides M13 5′ dT5overhang-30bp_c, M13 3′ dT5overhang-30bp_c and M13-overhang-30bp_c were used to created 5 nt-long tail variants with 30 bp complementary regions). The oligonucleotides were labeled at the 3′ terminus with [α-$^{32}$P] dCTP (Hartmann Analytic) by terminal transferase (New England Biolabs) prior to annealing, according to standard procedures. Unincorporated nucleotides were removed using Micro BioSpin™ P-30 Gel Columns (Bio-Rad). The annealed substrate was purified using CHROMA SPIN™ + TE-200 Columns (TaKaRa) to remove unannealed oligonucleotides. Sequences of all oligonucleotides used for substrate preparation are listed in Supplementary Table S3.

### Electrophoretic mobility shift assays
Binding reactions (15 μl volume) were carried out in 25 mM Tris-acetate pH 7.5, 3 mM EDTA, 1 mM dithiothreitol (DTT), 100 μg/ml bovine serum albumin (BSA, New England Biolabs), and DNA substrate (1 nM, in molecules). 3 mM EDTA was replaced with 4 mM ATP (Sigma, A7699) or 4 mM ATP-γ-S (Biolog) and 1 mM magnesium acetate, where indicated. Proteins were added and incubated for 15 minutes on ice. Loading dye (5 μl; 50% glycerol [w/vol] and bromophenol blue) was added to the reactions and the products were separated on 4% polyacrylamide gels (ratio acrylamide:bisacrylamide 19:1, Bio-Rad) in TAE (40 mM Tris, 20 mM acetic acid and 1 mM EDTA) buffer at 4 °C. The gels were dried on 17 CHR paper (Whatman), exposed to a storage phosphor screen (GE Healthcare) and scanned by a Typhoon Phosphor Imager (FLA9500, GE Healthcare). Signals were quantified using ImageJ and plotted with GraphPad Prism. DNA binding experiments using plasmid-length substrates were performed under the same conditions except with 100 ng DNA per reaction and were run on 0.8% agarose gel at 4 °C and post-stained with GelRed (Biotium).

### Helicase assays
Helicase assays (15 μl volume) were performed in a reaction buffer (25 mM HEPES-NaOH pH 7.5, 1 mM magnesium acetate, 4 mM ATP, 1 mM DTT, 0.1 mg/ml BSA) with DNA substrate (0.1 nM, in molecules). Where indicated, reactions contained 1 mM ATP and 5 mM magnesium acetate. Recombinant proteins were added as indicated. The reactions were incubated at 37 °C for 30 minutes and stopped by adding 5 μl 2% stop buffer (2% sodium dodecyl sulfate [SDS] [w/vol], 150 mM EDTA, 30% glycerol [w/vol] and bromophenol blue) and 1 μl of proteinase K (14−22 mg/ml, Roche) and incubating at 37 °C for 10 minutes. To avoid re-annealing of the substrate, the stop solution was supplemented with a 20-fold excess of the unlabeled oligonucleotide with the same sequence as the $^{32}$P-labeled one. The products were separated by 10% polyacrylamide gel electrophoresis in TBE (89 mM Tris, 89 mM boric acid, 2 mM EDTA) buffer, dried on 17 CHR paper and analyzed as described above.

### ATPase assays
ATPase assays with MCM8-9 and HROB were performed in reaction buffer (25 mM HEPES-NaOH pH 7.5, 5 mM magnesium acetate, 1 mM DTT, 0.1 mg/ml BSA, 0.1 mM ATP, 1 nM of [γ-$^{32}$P] ATP [Hartmann-Analytic]) with DNA as a co-factor (1 μM, in nucleotides, where indicated) in 10 μl reaction volume. Recombinant proteins were added as indicated. The samples were mixed on ice and incubated at 37 °C for 2 hours. Reactions were stopped with 2 μl of 0.5 M EDTA and separated using thin-layer chromatography plates (Merck) with 0.3 M LiCl and 0.3 M formic acid as the mobile phase. Dried plates were exposed to storage phosphor screens (GE Healthcare) and scanned by a Typhoon FLA 9500 phosphorimager (GE Healthcare). Signals were quantified using

ImageJ software. Spontaneous ATP hydrolysis signal from no protein lanes was removed as a background during quantitation. The percentage of ATP hydrolysis was obtained as a normalized value and expressed in arbitrary units, or as a rate of ATP hydrolysis.

## Mass photometry assays

Mass photometry measurements were performed on a 2MP-0132 mass photometer (Refeyn). For the measurements, coverslips (No. 1.5 H thickness, 24 × 50 mm, VWR) were cleaned by sequential dipping in Milli-Q-water, isopropanol and Milli-Q-water followed by drying under a stream of gaseous nitrogen. Subsequently, silicone gaskets (CultureWell™ Reusable Gasket, Grace Bio-Labs) were placed on the cleaned coverslips to create wells for sample loading. For mass measurements, gaskets were filled with 18 μl equilibration buffer (50 mM Tris–HCl pH 7.5, 100 mM NaCl) to allow focusing the microscope onto the coverslip surface. Subsequently, 2 μl protein solution was added into the 18 μl droplets and mixed to obtain 50-200 nM final protein concentration. Sample binding to the coverslip surface was monitored by recording a movie for 1 minute using AcquireMP (Refeyn Ltd) software. Data analysis was performed using DiscoverMP (Refeyn Ltd). For each peak analyzed, at least 500 events were considered. To convert the measured optical reflection-interference contrast into a molecular mass, a known protein size marker (NativeMark™ Unstained Protein Standard, Invitrogen) was measured. To visualize the interaction between MCM8-9 and HROB, MCM8-9 was preincubated with HROB (both at 1 μM) for 10 minutes at room temperature in the reaction buffer (25 mM Tris-HCl pH 7.5, 3 mM EDTA, 50 mM NaCl). The mix was then rapidly diluted to 200 nM concentration for measurement. To measure the oligomeric state of MCM8-9 heterodimer in the presence of DNA and ATP, MCM8 (R309D)-9 (5 μM, dimeric mutant) was incubated with 70 nt ssDNA (5 μM, in molecules) in binding buffer (25 mM HEPES-NaOH pH 7.5, 2 mM magnesium acetate, 1 mM ATP) for 15 minutes at room temperature. Reactions without ATP or DNA were used as controls.

## In vitro interaction studies

To study the interaction between recombinant MCM8-9 and HROB, *Sf*9 cells were co-infected with MBP-MCM9 and FLAG-MCM8 baculoviruses. Cells were lysed, and MCM8-9 was immobilized on amylose resin and washed with wash buffer (50 mM Tris–HCl pH 7.5, 2 mM β-mercaptoethanol, 300 mM NaCl, 0.1% (v/v) NP40, 1 mM PMSF). Resin-bound MCM8-9 was then incubated with 1 μg of wild type or mutant HROB, diluted in binding buffer (50 mM Tris–HCl pH 7.5, 2 mM β-mercaptoethanol, 3 mM EDTA, 100 mM NaCl, 0.2 μg/μl BSA, 1 mM PMSF) for 1 hour at 4 °C with continuous rotation. The resin was washed 4 times with wash buffer containing 100 mM NaCl, proteins were eluted in wash buffer supplemented with 10 mM maltose and detected by western blotting[33]. As a negative control, HROB was incubated with the resin without the bait protein. For the experiment comparing phosphorylated and dephosphorylated variants of MCM8-9, pMCM8-9 (prepared as described in the purification section to preserve the phosphorylation status) was immobilized on amylose resin and was treated with λ-phosphatase for the dephosphorylated conditions, followed by three washes using the wash buffer, before incubation with wild type HROB. Reciprocal interaction assays that used HROB as a bait and MCM8-9 variants as prey were performed using HROB-FLAG-Strep construct. 1 μg of Anti-Strep-tag II antibody (Abcam, ab184224) was captured on 10 μl Protein G magnetic beads (Dynabeads, Invitrogen) by incubating at 4 °C for 1 hour with gentle rotation in 50 μl PBS-T (PBS with 0.1% Tween-20 [Sigma]). The beads were then washed 2 times on a magnetic rack with 150 μl PBS-T. The beads were mixed with 1 μg of recombinant HROB-FLAG-Strep (232 nM) in 60 μl immunoprecipitation buffer (25 mM Tris-HCl pH 7.5, 1 mM DTT, 3 mM EDTA, 0.20 μg/μl BSA, 50 mM NaCl) and incubated at 4 °C for 1 hour with gentle rotation. Beads were then washed 3

times with 150 μl wash buffer (25 mM Tris-HCl pH 7.5, 1 mM DTT, 3 mM EDTA, 80 mM NaCl, 0.05% Triton-X [Sigma]). Variants of MCM8-9 as indicated in the figures were then added to the beads in 60 μl immunoprecipitation buffer (25 mM Tris-HCl pH 7.5, 1 mM DTT, 3 mM EDTA, 0.20 μg/μl BSA, 50 mM NaCl) and incubated at 4 °C for 1 hour with gentle rotation. Beads were again washed 3 times with 150 μl wash buffer (25 mM Tris-HCl pH 7.5, 1 mM DTT, 3 mM EDTA, 80 mM NaCl, 0.05% Triton-X) and proteins were eluted by boiling the beads in SDS buffer (50 mM Tris-HCl pH 6.8, 1.6 % SDS, 100 mM DTT, 10% glycerol, 0.01% bromophenol blue) at 95 °C for 3 minutes. The eluate was separated on a SDS-PAGE gel and the proteins were detected by silver staining.

## Structural Modeling

Sequences of human HROB (Q8N3J3), MCM8 (Q9UJA3), and MCM9 (Q9NXL9) were retrieved from UniProt database[62] and used as input of mmseqs2 homology search program[63] to generate a multiple sequence alignment (MSA) against the UniRef30 clustered database. Special care was taken in selecting only orthologs from the MSA of the proteins. For this, homologs of MCM8 and MCM9 sharing less than 35% sequence identity with their respective query and those of HROB sharing less than 25% were discarded. Homologs with less than 50% of coverage of the aligned region of their query were also discarded. In every MSA, in case several homologs belonged to the same species, only the one sharing highest sequence identity to the query was kept. Full-length sequences of the orthologs were retrieved and re-aligned with mafft[64]. To model the structure of the MCM8-9 complex with and without HROB, the corresponding MSAs were concatenated. In the concatenated MSAs, when homologs of different subunits belonged to the same species, they were aligned in a paired manner otherwise they were left unpaired. Final concatenated MSAs of HROB-MCM8-9 contained 1247 sequences with 1888 positions and 222 paired sequences while that of MCM8-9 contained 1247 sequences of 1539 positions with 690 matching sequences. The model of MCM8-9 complex was calculated as a hexameric assembly using a stoichiometry of 3 for every subunit. Concatenated MSAs were used as inputs to generate 5 structural models for each of the HROB-MCM8-9 trimer and of the MCM8-9 hexamer using a local version of the ColabFold v1.5.2 interface[65] running 3 iterations of the Alphafold2 v2.3.1 algorithm[66] trained on the multimer dataset[67] on a local HPC equipped with NVIDIA Ampere A100 80Go GPU cards. The five top models of each of the complexes converged toward very similar conformations and obtained good confidence and quality scores with pLDDTs in the range [73, 74.9] and [76.6, 80.1], pTMscore in the range [0.752, 0.773] and [0.706, 0.78] and ipTMscore in the range [0.705, 0.735] and [0.678, 0.757] for HROB-MCM8-9 and MCM8-9 hexamer, respectively. The models with the highest ipTMscores for both complexes were relaxed using rosetta relax protocols[68] to remove steric clashes under constraints (std dev. of 2 Å for the interatomic distances) and were used for structural analysis. Molecular graphics and analyses were performed with UCSF ChimeraX[69].

## Plasmids for cellular experiments

For immunoprecipitation, pMSCV-FLAG-HA-eGFP, -MCM8 and -MCM9 constructs were used[33]. To generate expression constructs for MCM8 (L387E) and MCM9 (M45E) mutants, Gateway entry vectors pDONR223-MCM8 and pDONR223-MCM9[33] were subjected to site-directed mutagenesis by inverse PCR using primers listed in Supplementary Table S4. The mutant entry vectors were then recombined into pMSCV-FLAG-HA-DEST using LR clonase II (Thermo Fisher, 11791020).

## Immunoprecipitation

HEK293T were transiently transfected with pMSCV-FLAG-HA-MCM8, -MCM8 L387E, -MCM9, -MCM9 M45E or -GFP control (Transporter 5,

Polysciences 26008-5) and harvested 3 days later for HA-immunoprecipitation[33]. Briefly, cell pellets were resuspended in mammalian cell lysis buffer (MCLB) (50 mM Tris-HCl pH 7.5, 1% IGEPAL CA-630) supplemented with 150 mM NaCl, a protease inhibitor (Goldbio, GB-331) and phosphatase inhibitor cocktail (Sigma, 4906837001). Following incubation for 30 minutes at 4 °C, extracts were cleared by centrifugation and supernatants collected. The remaining pellets were subjected to a second round of extraction by resuspending them in MCLB supplemented with 500 mM NaCl and protease (Goldbio, GB-331) and phosphatase inhibitor cocktail (Sigma, 4906837001), and incubating them for an additional hour at 4 °C. After clearing these extracts by centrifugation, the NaCl concentration of the collected supernatants was adjusted to 150 mM with MCLB and combined with the supernatants from the first round of extraction. Combined lysates were then incubated with anti-HA agarose beads (Sigma, A2095) for 4 hours at 4 °C. After incubation, beads were washed four times in MCLB buffer supplemented with 150 mM NaCl, and then boiled in LDS (lithium dodecyl sulfate) sample buffer (Thermo Fisher, NP0008) supplemented with β-mercaptoethanol to elute the bound proteins.

### Phosphatase Assay

HEK293T cells from a near-confluent 10 cm dish were harvested, pelleted by centrifugation and resuspended in 500 μl lysis buffer (50 mM Tris-HCl pH 7.5, 150 mM NaCl, 1% IGEPAL CA-630) supplemented with protease inhibitor cocktail (Goldbio, GB-331). Following incubation on ice for 15 minutes, extracts were cleared by centrifugation. Phosphatase reactions (100 μl total volume) containing cleared lysates supplemented with $MnCl_2$ (1 mM final concentration), 100 units lambda phosphatase (NEB, P0753S), and/or phosphatase inhibitor cocktail (2x final concentration) (Sigma, 4906837001) were prepared on ice and subsequently incubated at 30 °C for 1 hour. Reactions were then boiled with LDS sample buffer (Thermo Fisher, NP0008) supplemented with β-mercaptoethanol, resolved by SDS-PAGE using a 6% acrylamide gel supplemented with 75 μM PhosBind reagent (APExBIO, F4002) and 150 μM $MnCl_2$, and transferred to nitrocellulose membrane for immunoblotting.

### Antibodies

The following antibodies were used for immunoblotting: rabbit anti-HROB (Sigma HPA023393; 1:1,000 dilution), rabbit anti-MCM8 (Proteintech 16451-1-AP; 1:5,000 dilution), rabbit anti-MCM9 (EMD/Millipore ABE2603; 1:20000 dilution), mouse anti-HA (Sigma H3663; 1:2,500 dilution), mouse anti-vinculin (Sigma V9131; 1:20,000 dilution). For in vitro interaction studies, the following antibodies were used for western blotting: rabbit anti-MCM9 (Millipore ABE2603; 1:10,000 dilution) and mouse anti-FLAG (Sigma F1804; 1:1,000 dilution) for FLAG-MCM8 and HROB-FLAG wild type and mutants.

### Negative staining transmission electron microscopy

The stock of purified MCM8-9 complexes in purification/storage buffer was diluted to a final concentration of 220 nM in cold EM buffer (20 mM HEPES-NaOH pH 7.55, 130 mM NaCl, 2% glycerol, 0.5 mM DTT) on ice. Of this dilution, 5 μl were incubated for 1 minute at room temperature on a carbon film 300 mesh copper grid (CF300-CU from Electron Microscopy Sciences) that had been negatively glow discharged for 45 s at 25 mA using the Emitech K100X system. Using Whatman filter paper, the excess sample on the grid was blotted away and the grid washed twice in EM buffer. Subsequently, the grid was stained with two droplets of 2% uranyl acetate and air-dried. TEM imaging was performed using a FEI Morgagni 268 microscope with 100 kV and a CCD 1376 ×1032 pixel camera at several magnifications. The diameter of 31 clearly distinguishable MCM8-9 rings was measured in the TEM micrographs using the Morgagni User Interface 3.0 and iTEM 5.2 software, and the data was analysed using GraphPad Prism.

### Flow-cell preparation for magnetic tweezers

In preparation for the assembly of the flow-cell, two 60 mm×24 mm cover slides (Menzel, ThermoScientific) were soaked subsequently in ultrapure water, acetone and isopropanol for 10 minutes each. Following each cleaning step, the slides were dried with nitrogen gas. The top slide was additionally modified with two holes of a diameter of ~ mm serving as an inlet and an outlet. The bottom slide was coated with a 1% w/vol polystyrene-toluene solution via spin-coating at 6,000 rpm. To conclude the coating procedure, the bottom slide was placed in an oven for 90 minutes at 150 °C. The assembly of the flow-cell was finalized by placing a specifically cut out Parafilm layer in-between the top and the bottom slide, forming a cavity for the liquid flow. This arrangement was placed on a heating plate at 120 °C, thereby melting the Parafilm and permanently fusing the two slides together.

In order to allow for specific surface binding, the assembled flow cell was incubated for 24 hours with a 50 μg/ml solution of anti-digoxigenin in phosphate-buffered saline (PBS). Following this step, the anti-digoxigenin was replaced with a 20 mg/ml BSA solution in PBS and incubated for 24 hours. Next, the flow-cell was installed inside the magnetic tweezer setup and connected to a pump. The flow-cell was first flushed with PBS to remove excess BSA. Then, a solution of polystyrene beads with a size of 3.6 μm was flushed inside and left to incubate overnight, ensuring their firm attachment to the bottom slide of the flow-cell. These polystyrene beads will function as reference points for the subsequent magnetic-tweezer measurements. Prior to a magnetic tweezer measurement, 2.5 μl of streptavidin coated magnetic beads with a diameter of 1 μm (Dynabeads, MyOne, Invitrogen) were washed three times with PBS. The beads were then diluted in 2.5 μl PBS to which 1 μl of 40 ng/μl of Holliday junction construct was added. The mixture was left to incubate for 5 minutes at room temperature. Lastly, the incubated DNA-construct/magnetic bead mixture was resuspended in 100 μl PBS.

### HJ-containing DNA construct for magnetic tweezers

The Holliday junction construct was created from 4 parts: two linear DNA PCR fragments with lengths of 5,326 bp (long fragment) and 4,229 bp (short fragment), a biotin-modified DNA handle as well as a digoxigenin-modified DNA handle. The linear long and short fragments were synthesized via PCR from the plasmid pBluescript+1 + 2 + 4 (sequence available upon request) using the following primers: HJ-fragment 5326 bp Forward Primer, HJ-fragment 5326 bp Reverse Primer, HJ-fragment 4229 bp Forward Primer and HJ-fragment 4229 bp Reverse Primer. The fragments were subsequently digested with Nt.BbvCI and PspOMI (long fragment) or Nt.BbvCI and XhoI (short fragment), resulting in a mutually complementary 9 nt 3'-overhang of the two fragments. To ensure specific binding of the construct to the magnetic bead and flow-cell surface, respectively, ~550 bp biotin- and digoxigenin-modified DNA handles were produced by PCR from plasmid pBluescript II SK+ using biotin-modified dUTPs for the biotin handle and digoxigenin-modified dUTPs for digoxigenin handle[70]. The primers Handles Forward Primer and Handles Reverse Primer were used for both handles in pairs. Sequences of all used primers are listed in Supplementary Table S3. The biotin handle was digested with PspOMI and the digoxigenin handle with XhoI. The four distinct fragments (biotin-handle + long fragment + short fragment + digoxigenin handle) were then ligated in a single reaction, resulting in the 9546 bp Holliday junction construct (contour length ~3.1 μm). Subsequently, the construct was gel-purified and stored at -20 °C. Sequences of all used primers are listed in Supplementary Table S3.

### Magnetic tweezers measurements on HJ substrate

The DNA-construct-magnetic bead mixture was flushed into the flow-cell and incubated for 150 seconds. Afterwards, DNA molecules that did not specifically bind to the flow-cell surface were removed by washing with PBS. A suitable field of view, which included a stable reference bead and a magnetic bead tethered to the flow-cell surface

by the DNA, was selected. To validate that the chosen construct was able to extrude the desired Holliday junction, it was examined if it was torsionally constrained. For that, 40 positive turns with respect to the helicity of the stretched dsDNA were applied at a force of 3.5 pN. Then, the force was lowered to 1 pN. If the result of this reduction in exerted force was a drastic change in the observable length of the DNA, the construct was confirmed as being torsionally constrained. When this prerequisite was met, 400 negative turns with respect to the helicity of the stretched dsDNA were applied at a force of 3.5 pN. The generated torque was sufficient to melt the hydrogen bonds of the DNA. Then the applied force was lowered to 1 pN in order to stimulate the formation of the Holliday junction. The incorporated mismatch in the center of the palindromic sequence served as an accessible starting point. After 1 hour, the formation of the junction could be observed as a shortening in DNA length. After successful formation of the Holliday junction, the total amount of negative turns was reduced to 50, in order to lower the torsional stiffness, thereby, making it easier for the investigated proteins to unwind the junction. PBS buffer was exchanged with the MCM buffer (25 mM HEPES-NaOH, 1 mM magnesium acetate, 1 mM DTT, 0.1 mg/ml BSA and 4 mM ATP) before measuring the proteins. Then, a force of 10 pN was applied, reducing the fluctuations of the magnetic bead to a minimum of around ± 9 nm. For each measurement, several baseline traces for the behavior of the junction in the MCM-buffer alone were recorded. The proteins were pre-mixed at the indicated concentrations and then flushed into the flow-cell, and the traces were recorded subsequently. Magnetic tweezers single molecule data were collected using Labview 2016 and CUDA software. Data analysis as well as plotting of magnetic tweezers data was carried out in Origin Lab 2019.

### Reporting summary

Further information on research design is available in the Nature Portfolio Reporting Summary linked to this article.

## Data availability

The structural models are available in ModelArchive (modelarchive.org) with the accession codes ma-ji5l4 and ma-zlpye for the MCM8-MCM9 hexamer and HROB-MCM8-MCM9 complex, respectively. Movies underlying mass photometry analysis and single-molecule source data are uploaded to Dryad [https://doi.org/10.5061/dryad.wdbrv15wq]. The previously published experimental structure of human ADP-bound MCM8-9 8S91 was used as a reference. Source data are provided with this paper.

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

## Acknowledgements

We thank members of the Cejka Lab for their critical comments on this
manuscript. We thank Boris Pfander (Max-Planck Institute of Biochem-
istry, Germany) for the CDK2-CycA expression construct and Roshan
Singh Thakur (Protein facility, Institute for Research in Biomedicine,
Bellinzona) for assistance with size exclusion chromatography. The
Swiss National Science Foundation (SNSF) (Grants 310030_207588 and
310030_205199) and the European Research Council (ERC) (Grant
101018257) support the research in the PC laboratory. The French
National Research Agency (ANR) (Grant ANR-21-CE44-0009-01) support
the RG laboratory. The work in the Ciccia laboratory was supported by
the NIH grant R01CA197774. The European Research Council under the
European Union's Horizon 2020 research and innovation programme
(ERC-2016-CoG-724863) and the European Social Fund (ESF) and the
Free State of Saxony (Junior Research Group UniDyn, Project No. SAB
100382164) to RS laboratory also supported this work.

## Author contributions

A.A., R.G. and P.C. conceived the idea. A.A. designed and performed
biochemical ensemble and mass photometry experiments. M.M. and
M.G performed the magnetic tweezer experiments under the super-
vision of R.S. H.B. and R.G. performed modeling experiments and
mutation design. V.K. performed the TEM experiments. J.W.H. per-
formed the experiments with cell extracts under the supervision of A.C.

A.A. and P.C. wrote the manuscript. All authors contributed to prepare the final version of the manuscript.

## Competing interests

The authors declare no competing interests.
