## [Peer Review File · Nature Communications]

Mechanism of DNA unwinding by MCM8-9 in complex with HROBREVIEWER COMMENTS

Reviewer #1 (Remarks to the Author):

This manuscript uses AlphaFold2 to predict the structure of the MCM8-9 heterodimer and hexamer in association with HROB and to identify interacting residues at the interfaces to establish function. The results are generally of high quality. New information presented are:

1. HROB acts downstream of MCM8-MCM9 loading to facilitate ATPase activity and promotes translocation and productive unwinding activity
2. HROB interacts with both Mcm8 and Mcm9 in the neck region verified by mutation analysis.
3. Low DNA unwinding processivity based on single molecule experiment, HJ is the preferred substrate
4. MCM8-9 unwinds DNA as a hexameric complex that assembles from dimers in the presence of ATP
5. Two distinct interfaces between alternating MCM8 and MCM9 subunits. Interface I, dimerization; interface II, hexamerization, ATPase and helicase activity

This is a very well-written and informative study.

Concerns that need to be addressed.

1. What is the stoichiometry of HROB:MCM8-9? Does concentration of HROB:MCM8-9 affect processivity? Would cross-linking the HROB:MCM8-9 complex using gravix help determine the stoichiometry?
2. Extended fig 2g, is nM refer to aa or MW of HROB and MCM8-9? This should be pointed out in the figure legend.
3. p. 8 Not clear how polarity of translocation is determined. Should determine polarity instead of referencing studies using yeast MCM2-7 (51) and archeal MCM (52). There is some inconsistency in the M13 and oligonucleotide substrates.... Speculations should be moved to discussions.
4. Authors referred to a paper in press in eLife that reports the cryoEM structure of MCM8-9 which is consistent with this study using AlphaFold2. It would indeed be interesting to compare the similarities and differences in the two approaches and could serve as a benchmark for future structural analyses based on AlphaFold2. It would be good for the authors to discuss what additional information that could have been obtained from cryoEM studies that cannot be obtained by AlphaFold2.
5. Does this MCM8-9 paper in press in eLife include structure of HROB complexed with MCM8-9? Are the results consistent?

Reviewer #2 (Remarks to the Author):

General comment

In the manuscript 'Mechanism of DNA unwinding by hexameric MCM8-9 in complex with HROB', Acharya et al used molecular modeling, biochemical assays, and single-molecule analysis to investigate the mechanism of DNA unwinding by MCM8-9 and its regulatory factor HROB. The purified MCM8-9 exists mainly in a dimeric state, and is assembled onto ssDNA to form MCM8-9 hexamer dependently on ATP, and then HROB promotes the DNA unwinding activity of MCM8-9 hexamer. The authors defined two distinctive interfaces between MCM8 and MCM9: Interface I contributes to the formation of MCM8-9 dimer, while Interface II dynamically combines the dimer to build the hexamer. These findings are interesting, but I have several concerns which should be addressed for this work to be published in Nature Communications. The major point is a lack of direct evidence for MCM8-9 hexamer formation on substrate DNAs. As it is likely that the enhanced binding of MCM8-9 on ssDNA in the presence of ATP reflects the hexamer formation, it will be worth examining the DNA binding abilities of various MCM8 and MCM9 mutants in addition to DNA unwinding activities.

Specific comments

1. In p.3, 'These proteins contain an N-terminal oligonucleotide-oligosaccharide (OB-fold) domain, which is ---' should be changed into '--- an N-terminal oligonucleotide-oligosaccharide-binding (OB-fold) domain, ---'
2. In Figure 2a-d (& Expanded Data Fire 2a-f), the authors showed that CDK-dependent phosphorylation of MCM8-9 suppressed DNA unwinding activity. However, this part seems isolated from other parts and to interrupt the main storyline. I think this part may be deleted or transferred to Expanded Data Figure.
3. In Expanded Data Figure 2h, it may be better to designate the position (5' or 3') of the 32P-labeling on oligonucleotides. In addition, MCM8-9 showed comparatively high unwinding activity for a 5'-overhang substrate without HROB only in this experiment. Are there any reasons?
4. The abilities of HROB to bind linear ssDNA oligonucleotides seems quite different between Figure 1C and Figure 2h. Is there any difference in experimental conditions?
5. Although the magnetic tweezer technique in Figure 3 and Expanded Data Figure 3 is a powerful tool, these experiments only showed a low processivity for DNA unwinding by MCM8-9. The branch migration activity was detected, but further experiments using Holliday junction were not performed in the later Figures. The presence of this part seems to disconnect a linkage between Figure 2 i-k (ATPase activity) and Figure 4 (DNA unwinding activity of ATPase mutants). Therefore, the whole part may be deleted or transferred to Expanded Data Figure.
6. In Figure 5, the binding of dimer-rich MCM8-9 to circular ssDNA was stabilized in the presence of ATP. This was not observed when using MCM8-K460A or MCM9-K358A mutants. Have the authors ever examined the effect of ATP- γ -S in this experiment? Although the DNA unwinding activity of MCM8-9 complex containing MCM8-K460A was severely impaired, but that of MCM9-K358A containing complex was only moderately affected (Figure 4f, Figure 6e). Therefore, it is possible that ATP binding (not ATP hydrolysis) is sufficient for the enhanced binding of MCM8-9 to circular ssDNA (or the formation of MCM8-9 ring on ssDNA). In relation with this, it may be interesting to examine the effect of ATP- γ -S for the oligomeric state of MCM8-9 by mass photometry analysis.
7. With relation to Figure 6 and Expanded Data Figure 6, do the authors have any information about DNA binding ability of MCM8-9 interface II mutants, especially MCM8-R309D & MCM9 -D230R? These mutants did not form hexamer in solution, but showed high DNA unwinding activity (Fig. 6c, 6d). Therefore, these mutants should show the enhanced binding to ssDNA if it really reflects the formation of MCM8-9 ring.
8. In Figure 6e, it is unclear whether the dimeric ATPase (MCM8-WT/MCM9-WT, MCM8-WT/MCM9-K358A, MCM8-K460A/MCM9-WT) was combined with monomeric interface II mutants (MCM8-R309D, MCM9-D230R) or dimeric interface II mutants (MCM8-R309D/MCM9-WT, MCM8-WT/MCM9-D230R). The result of Fig. 6e indicates that the presence of MCM8-R309D or MCM9-D230R stimulated the DNA unwinding activity when combined with wild type dimeric MCM8-9, but unchanged it when combined with MCM8-WT/MCM9-K358A. This may be due to defective stabilization of MCM8-9 on DNA. Therefore, it may be worth examining the DNA binding ability of these combinations of MCM8-9 dimers.
9. Is it possible to show the hexamer formation on linear ssDNA in the presence of ATP using mass photometry? Although EMSA can only detect MCM8-9 rings topologically locked on circular DNA, mass photometry may be able to detect MCM8-9 hexamer rings slid of the ends of linear DNA.
10. If possible, it will be worth examining the oligomeric state of endogenous MCM8-9 complex in cells (mainly in a dimeric?).
11. In Discussion, the authors speculated about the dependency of MCM8-9 complex on HROB for their recruitment onto chromatin. The data of Figure 2 and 3 shows that HJ is the most preferable substrate for MCM8-9 and that HROB has significantly higher affinity for HJ than for Y. The binding of MCM8-9 to HJ may be enhanced or stabilized by HROB, differently from the binding of MCM8-9 to ssDNA, which was not enhanced by HROB.

Reviewer #3 (Remarks to the Author):

In this study, the authors employed multiple approaches to investigate the function of MCM8-9 helicase and its regulation by HROB. Molecular modeling was used to identify key residues from both HROB and MCM8-9 that are essential for HROB binding to MCM8-9. The authors showed that these interactions are crucial for stimulating MCM8-9's helicase activity. The authors also proposed that HROB binding activates the ATPase activity of MCM8-9, although the evidence for this claim is limited. The study further examined the activity of MCM8-9 on various DNA substrates and discovered that the MCM8-9 helicase favors Holliday Junction (HJ) and Y-shaped DNA templates. Overall, these findings offer novel insights into the function of MCM8-9 helicase, which would be of interest for the fields of DNA replication, damage repair and related areas. However, despite the reviewer's enthusiasm for this work, the present manuscript has various issues and concerns that need addressing.

Major concerns:

1. The authors need to provide concrete evidence to substantiate their claim in the manuscript's title that MCM8-9 operates as a hexamer for DNA unwinding through forming a complex with HROB.
2. The authors found that the OB domain of HROB plays an important role in stimulating the helicase activity of MCM8-9. However, It is not clear how the removal of the OB domain affects the affinity of HROB binding to MCM8-9. This should be addressed.
3. The authors used molecular modeling to predict the structures of MCM8-9 hexamer with and without HROB. However, since the structures of related MCM8-9 structures, including a full-length chicken MCM8-9 (gMCM8-9, PDB: 7W7P; EMD-32346) and the NTD ring of human MCM8-9 (hMCM8-9) with and without HROB bound (Li et al 2021, Structure; Weng et al 2023, eLife) have been determined, it is important to compare the predicted structures with the available real ones to assess the reliability of the predications.
4. The recent structure of hMCM8-9-NTD bound with HROB (Weng et al 2023. eLife) revealed conformational changes in the NTD ring of hMCM8-9; however, the HROB is not visible. This observation suggests that the flexible HROB may have multiple weak and/or transient binding surfaces on MCM8-9, similar to the structure of CMG bound with Mcm10 (Mayle et al 2018 PNAS; Yuan et al 2020 Nat Commun). Therefore, the authors should validate the predicted structure. Concrete evidence should be provided to demonstrate whether HROB binds to MCM8-9 hexamer. If the assembly is unstable, crosslinking should be applied for fixation. The relevant MCM8-9-HROB should be isolated for structure determination using cryo-EM and cross-linking mass spectrometry analysis. These analyses will provide concrete structural information, elevating the quality of this paper to a level suitable for publication in Nature Communications.
5. The authors claim that HROB-F553E mutant loses the interaction with MCM8-9. This mutation site is in the OB domain of HROB. However, previous results from the same research team showed that the OB domain of HROB is dispensable for its binding to MCM8-9 (Huang et al 2020 Nat Commun). HROB-1-432 mutant, in which the OB domain is removed, can pull down MCM8-9. Therefore, it is unclear how a single point mutation, F553E, completely abolish the binding of HROB to MCM8-9 as shown in Extended Fig. 1c. This needs to be addressed.
6. The authors found that ATP hydrolysis by MCM8-9 is only slightly enhanced by HROB. However, HROB induces a significant increase in the activity of MCM8-9 in DNA unwinding. It is important to address this observation to establish a possible mechanism by which HROB regulates the function of MCM8-9 in DNA unwinding.
7. In addition, despite the presence of HROB, the authors observed low processivity of MCM8-9 in DNA unwinding. The authors should discuss how this property of MCM8-9 might be applied in homologous recombination?

RESPONSE TO REVIEWER COMMENTS

NCOMMS-23-26863-T by Acharya et al.

November 21, 2023

We would like to thank the reviewers for their time reading the manuscript and for providing helpful comments and suggestions. Below we provide point-by-point responses.

Reviewer #1 (Remarks to the Author):

This manuscript uses AlphaFold2 to predict the structure of the MCM8-9 heterodimer and hexamer in association with HROB and to identify interacting residues at the interfaces to establish function. The results are generally of high quality. New information presented are:

- 1. HROB acts downstream of MCM8-MCM9 loading to facilitate ATPase activity and promotes translocation and productive unwinding activity*
- 2. HROB interacts with both Mcm8 and Mcm9 in the neck region verified by mutation analysis.*
- 3. Low DNA unwinding processivity based on single molecule experiment, HJ is the preferred substrate*
- 4. MCM8-9 unwinds DNA as a hexameric complex that assembles from dimers in the presence of ATP*
- 5. Two distinct interfaces between alternating MCM8 and MCM9 subunits. Interface I, dimerization; interface II, hexamerization, ATPase and helicase activity*

This is a very well-written and informative study.

Concerns that need to be addressed.

- 1. What is the stoichiometry of HROB:MCM8-9? Does concentration of HROB:MCM8-9 affect processivity? Would cross-linking the HROB:MCM8-9 complex using gravix help determine the stoichiometry?*

RESPONSE: MCM8-9 and HROB do not form a stable complex. Whereas the MCM8 and MCM9 proteins can be co-expressed and co-purified as a stable 1:1 heterodimer, it is not the case with HROB. In the previous version of the manuscript, we only used Western blotting to detect the MCM8-9 and HROB interactions in pulldown assays, because we always brought down substoichiometric amounts.

In the revised version, we present mass photometry experiments, showing that when the proteins are pre-incubated at a high concentration (1 μ M), and quickly diluted to 200 nM (upper end for the instrument), we can detect MCM8-9 and HROB complex at 1:1 ratio, showing that one HROB molecule binds one MCM8-9 heterodimer (new Fig. 1a-c). When the proteins were mixed at lower concentrations (100 nM), the complex was not visible (new Supplementary Fig. 1b), reflecting the low affinity or transient nature of the interaction. The mass photometry data presented in Fig. 1c are in agreement with our AlphaFold2 model depicting HROB binding across the stable MCM8-9 heterodimer interface (Fig. 1i and 6f). As we could not experimentally detect the stoichiometric complex, we did not perform crosslinking experiments.

Functionally, we noted that HROB stimulates the MCM8-9 helicase activity best at sub-stoichiometric amounts (1:2 or 1:3 ratio). We believe the reason for that might be somewhat artefactual, not reflecting the binding stoichiometry. HROB that is not in complex with MCM8-9 binds DNA and non-specifically inhibits unwinding, as demonstrated by experiments with the BLM helicase. Keeping $[HROB] < [MCM8-9]$ then simply reduces the free concentration of HROB in the assay. We believe the point is clear in revised text. In response to the reviewer question, we also performed bulk assays and single molecule helicase assays that show that the $[HROB] : [MCM8-9]$ ratio does not affect processivity of the helicase ensemble. These data are included in new Supplementary Figures 5a and 6a-f.

- 2. Extended fig 2g, is nM refer to aa or MW of HROB and MCM8-9? This should be pointed out in the figure legend.*

RESPONSE: We now list protein concentrations (nM) used in pulldown assays in legends throughout the manuscript.

3. p. 8 Not clear how polarity of translocation is determined. Should determine polarity instead of referencing studies using yeast MCM2-7 (51) and archeal MCM (52). There is some inconsistency in the M13 and oligonucleotide substrates.... Speculations should be moved to discussions.

RESPONSE: The helicase ensemble does not unwind simple overhanged substrates, which are typically used to determine polarity. We note that a short non-complementary flap, forming a Y-junction at the ssDNA-dsDNA junction point is necessary for unwinding. The interpretation of those experiments to determine polarity was however tricky, as it was ambiguous on which strand of the Y junction MCM8-9 translocated. In the revised manuscript, we have replaced the previous M13 based experiments with new ones, where we notably reduced the length of the flaps, which do not allow the loading of MCM8-9. These experiments allowed us to conclude that the polarity is 3'-5', in accord with PMID: 37309874. Please refer to new Fig. 2e,f and the related text for more details.

4. Authors referred to a paper in press in eLife that reports the cryoEM structure of MCM8-9 which is consistent with this study using AlphaFold2. It would indeed be interesting to compare the similarities and differences in the two approaches and could serve as a benchmark for future structural analyses based on AlphaFold2. It would be good for the authors to discuss what additional information that could have been obtained from cryoEM studies that cannot be obtained by AlphaFold2.

We have compared the models of the MCM8-9 hetero-hexamers generated with AlphaFold2 with two recently deposited structures corresponding to the helicase structures of human MCM8-MCM9 in apo form (eLife, PMID: 37535404) (PDB: 7WI7) or in ADP-bound with DNA conditions (NAR, PMID: 37309874) (PDB: 8S91).

Generally, the structures are very similar, particularly with respect to PDB: 8S91. We began by comparing the structures of the N-ter and C-ter hexameric rings independently, to avoid questions of inter-ring mobility. For the N-ter and C-ter rings independently, there is more difference between the apo and the ADP-bound structures than between the ADP-bound structure and the AF2 model (Table R1 and R2). RMSD values of about 2 Å between the AlphaFold2 models of the two rings and the experimental cryoEM structure 8S91 highlight the general reliability of the modelled assembly (new Supplementary Fig. 11a). When comparing the relative orientations between the N-terminal and C-terminal domains of either MCM8 or MCM9, we observe a much larger variability (Table R3 and R4). This variability exists between the two experimental structures with RMSD values ranging from 3.5 to 4.5 Å. This variability arises from different combinations of rotations of about 10° and translations by up to 3 Å between the N- and C- rings (new Supplementary Fig. 11b).

Given the size of the two-rings, these moves can result in significant atomic displacement at the tips of the loops involved in DNA binding either in the C-terminal or N-terminal domains. Therefore, it is conceivable that the conformational states in which the AF2 models converged could be representative of conformations explored during the functional cycle of the enzyme, at least at the level of a single subunit. In the conformation generated by AlphaFold2 for the MCM8-9 hexamer, the N-ter and C-ter rings are closer together than in the two experimental structures. A more compact conformation might rather represent an ATP- than an ADP-bound state, since binding to both ATP and DNA was associated with smaller inter-ring distances in the structures of the drosophila MCM2-7 helicase (PMID: 31484077).

Importantly, the structures do not contain visible DNA, and HROB was *either* not included (PMID: 37309874) or was not visible (PMID: 37535404) in the structure. Therefore, AlphaFold2 allowed us to model these transient interactions and helped to go further with respect to the structural studies. We have designed mutants and employed them in biochemical experiments to validate the models.

5. Does this MCM8-9 paper in press in eLife include structure of HROB complexed with MCM8-9? Are the results consistent?

The authors attempted to obtain a structure of HROB-CTD and hMCM8/9 NTD, however they note they could not observe a density for HROB-CTD.

RMSD	M89hexa_Nter (ADP, 8S91)	M89hexa_Nter (apo, 7W17)
M89hexa_Nter (AF2 model)	1.9 Å (1980 aa)	3.9 Å (1662 aa)
M89hexa_Nter (ADP, 8S91)	-	4.1 Å (1608 aa)

Table R1: RMSD (Root Mean Square Deviation) between the 3 structures AF2 model, 7W17 and 8S91 of the MCM8 and MCM9 hexameric assembly focusing on the N-terminal domains forming ring (MCM8 66-365, MCM9 1-271).

RMSD	M89hexa_Cter (ADP, 8S91)	M89hexa_Cter (apo, 7W17)
M89hexa_Cter (AF2 model)	2.2 Å (1265 aa)	2.5 Å (1283 aa)
M89hexa_Cter (ADP, 8S91)	-	3.5 Å (1259 aa)

Table R2: RMSD (Root Mean Square Deviation) between the 3 structures (AF2 model, 7W17 and 8S91) of the MCM8 and MCM9 hexameric assembly focusing on the C-terminal domains forming ring (MCM8 400-840, MCM9 298-665).

RMSD	M8_Nter+Cter (ADP, 8S91)	M8_Nter+Cter (apo, 7W17)	M8_Nter+Cter (AF2 + HROB)
M8_Nter+Cter (AF2 model)	3.5 Å (602 aa)	4.3 Å (565 aa)	3 Å (775 aa)
M8_Nter+Cter (ADP, 8S91)	-	4.2 Å (545 aa)	3.9 Å (602 aa)
M8_Nter+Cter (AF2 + HROB)	-	3.6 Å (565 aa)	-

Table R3: RMSD (Root Mean Square Deviation) between the 4 structures of the MCM8 subunit (66-840) in the different assemblies corresponding to the AF2 model with HROB (in the hetero-trimer) and without (in the hexamer), 7W17 and 8S91.

RMSD	M9_Nter+Cter (ADP, 8S91)	M9_Nter+Cter (apo, 7W17)	M9_Nter+Cter (AF2 + HROB)
M9_Nter+Cter (AF2 model)	3.7 Å (621 aa)	3.6 Å (566 aa)	2.9 Å (665 aa)
M9_Nter+Cter (ADP, 8S91)	-	3.4 Å (540 aa)	3.9 Å (621 aa)
M9_Nter+Cter (AF2 + HROB)	-	4.3 Å (566 aa)	-

Table R4: RMSD (Root Mean Square Deviation) between the 4 structures of the MCM9 subunit (1-665) in the different assemblies corresponding to the AF2 model with HROB (in the hetero-trimer) and without (in the hexamer), 7W17 and 8S91.

Reviewer #2 (Remarks to the Author):

General comment

In the manuscript 'Mechanism of DNA unwinding by hexameric MCM8-9 in complex with HROB', Acharya et al used molecular modeling, biochemical assays, and single-molecule analysis to investigate the mechanism of DNA unwinding by MCM8-9 and its regulatory factor HROB. The purified MCM8-9 exists mainly in a dimeric state, and is assembled onto ssDNA to form MCM8-9 hexamer dependently on ATP, and then HROB promotes the DNA unwinding activity of MCM8-9 hexamer. The authors defined two distinctive interfaces between MCM8 and MCM9: Interface I contributes to the formation of MCM8-9 dimer, while Interface II dynamically combines the dimer to build the hexamer. These findings are interesting, but I have several concerns which should be addressed for this work to be published in Nature Communications. The major point is a lack of direct evidence for MCM8-9 hexamer formation on substrate DNAs. As it is likely that the enhanced binding of MCM8-9 on ssDNA in the presence of ATP reflects the hexamer formation, it will be worth examining the DNA binding abilities of various MCM8 and MCM9 mutants in addition to DNA unwinding activities.

Specific comments

1. In p.3, 'These proteins contain an N-terminal oligonucleotide-oligosaccharide (OB-fold) domain, which is ---' should be changed into '--- an N-terminal oligonucleotide-oligosaccharide-binding (OB-fold) domain, ---'

RESPONSE: Thank you. This is included and updated.

2. In Figure 2a-d (& Expanded Data Figure 2a-f), the authors showed that CDK-dependent phosphorylation of MCM8-9 suppressed DNA unwinding activity. However, this part seems isolated from other parts and to interrupt the main storyline. I think this part may be deleted or transferred to Expanded Data Figure.

RESPONSE: Yes, we agree and this part was moved to the supplementary data section. We have also removed the phosphorylation part from Discussion to streamline the text.

3. In Expanded Data Figure 2h, it may be better to designate the position (5' or 3') of the ³²P-labeling on oligonucleotides. In addition, MCM8-9 showed comparatively high unwinding activity for a 5'-overhang substrate without HROB only in this experiment. Are there any reasons?

RESPONSE: We now present clearer cartoons indicating the positions of the radioactive labels. With respect to the high activity of MCM8-9 without HROB for the 5' flap substrate: these experiments were done with fully phosphorylated MCM8-9 (prepared with phosphatase inhibitors), which is very active under no added salt conditions of the helicase assays. With HROB, the activity was only moderately enhanced because the assay was already almost saturated, leading to the misleading impression that HROB was not necessary for stimulation (if we used lower protein concentration, the stimulation would be very apparent). For consistency, we have replaced the experiments with those carried out with the regular (not hyperphosphorylated) MCM8-9. Additionally, as noted above in response to Reviewer #1, we altered the length of the overhangs to better determine the unwinding polarity.

4. The abilities of HROB to bind linear ssDNA oligonucleotides seems quite different between Figure 1C and Figure 2h. Is there any difference in experimental conditions?

RESPONSE: Thank you for pointing this out. The difference is primarily due to the length of the employed ssDNA substrates. Fig. 1c (now Fig. 1f) was done with 93 nt long ssDNA, while the experiment in Fig. 2h (now Fig. 2d) was carried out with shorter, 50 nt long ssDNA, which is bound less efficiently. We now show a panel (new Supplementary Fig. 1f) that compares the binding of HROB to ssDNA of various lengths (25, 50 and 93 nt). We also note that the OB fold of HROB does not notably change this DNA binding behavior. The Figure R1 shows the representative EMSAs (new Supplementary Fig. 1f only shows quantitations).

Figure R1: Electrophoretic mobility shift assay with human HROB or HROB-ΔOB and 25, 50 and 93 nt-long ssDNA. The red asterisk indicates the position of the radioactive label. Shown is a representative experiment.

5. Although the magnetic tweezer technique in Figure 3 and Expanded Data Figure 3 is a powerful tool, these experiments only showed a low processivity for DNA unwinding by MCM8-9. The branch migration activity was detected, but further experiments using Holliday junction were not performed in the later Figures. The presence of

this part seems to disconnect a linkage between Figure 2 i-k (ATPase activity) and Figure 4 (DNA unwinding activity of ATPase mutants). Therefore, the whole part may be deleted or transferred to Expanded Data Figure.

RESPONSE: The magnetic tweezer assay demonstrates that MCM8-9 and HROB ensemble has a low DNA unwinding processivity. We think that this is an important point to be made, in a paper focused on initial biochemical analysis, which extends the preceding Figures in the manuscript. Previously published data on the biochemistry of the MCM8-9 and HROB helicase complex are very limited, so we feel that the more general biochemical characterization (Figures 1-3) is warranted. Figures 3-6 then focus in more detail on the oligomerization of the complex. We have reorganized some of the panels in Figure 2, and attempted to improve the flow of the text.

6. In Figure 5, the binding of dimer-rich MCM8-9 to circular ssDNA was stabilized in the presence of ATP. This was not observed when using MCM8-K460A or MCM9-K358A mutants. Have the authors ever examined the effect of ATP- γ -S in this experiment? Although the DNA unwinding activity of MCM8-9 complex containing MCM8-K460A was severely impaired, but that of MCM9-K358A containing complex was only moderately affected (Figure 4f, Figure 6e). Therefore, it is possible that ATP binding (not ATP hydrolysis) is sufficient for the enhanced binding of MCM8-9 to circular ssDNA (or the formation of MCM8-9 ring on ssDNA). In relation with this, it may be interesting to examine the effect of ATP- γ -S for the oligomeric state of MCM8-9 by mass photometry analysis.

RESPONSE: The reviewer was right that ATP binding is sufficient for the enhanced binding of MCM8-9 to circular ssDNA. We have carried out the EMSA experiments with ATP- γ -S as suggested, and observed a similar stabilization as with ATP (new Supplementary Fig. 8b). Without DNA, the oligomeric state of MCM8-9 does not change even with ATP- γ -S (new Supplementary Fig. 7h,i), showing that the ring formation is DNA dependent.

7. With relation to Figure 6 and Expanded Data Figure 6, do the authors have any information about DNA binding ability of MCM8-9 interface II mutants, especially MCM8-R309D & MCM9-D230R? These mutants did not form hexamer in solution, but showed high DNA unwinding activity (Fig. 6c, 6d). Therefore, these mutants should show the enhanced binding to ssDNA if it really reflects the formation of MCM8-9 ring.

RESPONSE: The dimer mutants still bind DNA, even without ATP, and the EMSA experiments were not any more informative with these variants than with the wild type proteins. However, we used mass photometry and could observe that the hexamer species form only with ATP and with DNA using these mutants (new Supplementary Fig. 9c-f).

8. In Figure 6e, it is unclear whether the dimeric ATPase (MCM8-WT/MCM9-WT, MCM8-WT/MCM9-K358A, MCM8-K460A/MCM9-WT) was combined with monomeric interface II mutants (MCM8-R309D, MCM9-D230R) or dimeric interface II mutants (MCM8-R309D/MCM9-WT, MCM8-WT/MCM9-D230R). The result of Fig. 6e indicates that the presence of MCM8-R309D or MCM9-D230R stimulated the DNA unwinding activity when combined with wild type dimeric MCM8-9, but unchanged it when combined with MCM8-WT/MCM9-K358A. This may be due to defective stabilization of MCM8-9 on DNA. Therefore, it may be worth examining the DNA binding ability of these combinations of MCM8-9 dimers.

RESPONSE: We have realized that the labeling of this panel may have been unclear. The proteins used in the assay are not combinations of two sets of ATPase and interface II mutants, but rather single preps with multiple mutations. We have changed the labeling to make it clearer. The reviewer is likely correct to suggest that the lack of stimulation of K358A by the interface II mutants is likely due to productive hexamer destabilization caused by the combination of the two sets of mutations. However, the EMSA experiments can be ambiguous, due to what we believe is non-productive DNA binding without ATP, and we do not feel confident to make this point.

We use this assay to demonstrate that even if we partially impair the unstable interface II by the DR/RD mutations, it is the unstable interface II that is required to hydrolyze ATP for productive DNA unwinding, suggesting that the complex unwinds DNA as an oligomeric species higher than a dimer.

9. Is it possible to show the hexamer formation on linear ssDNA in the presence of ATP using mass photometry? Although EMSA can only detect MCM8-9 rings topologically locked on circular DNA, mass photometry may be able to detect MCM8-9 hexamer rings slid of the ends of linear DNA.

RESPONSE: We used linear ssDNA (70 nt long) and the dimeric MCM8 (R309D)-9 mutant in mass photometry. The protein alone without DNA and without ATP is exclusively dimeric. With DNA but without ATP, we also detect higher molecular weight complexes, likely corresponding to two MCM8-9 heterodimers independently bound to DNA. With DNA and ATP, we did observe species corresponding to the hexamer. We cannot clearly distinguish whether the hexamers are still bound to DNA or slid off the ends (Supplementary Fig. 9c-f).

10. If possible, it will be worth examining the oligomeric state of endogenous MCM8-9 complex in cells (mainly in a dimeric?).

RESPONSE: We prepared cell extracts from Expi293 cells, and treated them with benzonase to degrade DNA. We then performed analytical size exclusion chromatography and analyzed the fractions by Western blot as shown below in Figure R2. The data show that endogenous MCM8-9 migrates as a hexamer (or even higher) species. This is agreement with Nishimura et al. (PMID 22771115), where the authors also noted that chicken MCM8-9 likely forms a hexamer in cells. However, we formally cannot distinguish whether the migration on the size exclusion column reflects oligomer formation, or complex formation with other proteins (likely, a bit of both). We believe that this will be a useful assay to investigate the identified mutants in knockdown-rescue experiments in cells, but we would like to consider these assays above the scope of the current manuscript.

Figure R2: Size exclusion profile of the whole cell lysate (mixed with BSA as an internal marker) from Expi293 cells as opposed to the marker (thyroglobulin+BSA) and subsequent blotting for MCM8.

11. In Discussion, the authors speculated about the dependency of MCM8-9 complex on HROB for their recruitment onto chromatin. The data of Figure 2 and 3 shows that HJ is the most preferable substrate for MCM8-9 and that HROB has significantly higher affinity for HJ than for Y. The binding of MCM8-9 to HJ may be enhanced or stabilized by HROB, differently from the binding of MCM8-9 to ssDNA, which was not enhanced by HROB.

RESPONSE: We evaluated the binding of MCM8-9 and/or HROB to HJ (Fig. R3). As with ssDNA, we do see somewhat enhanced binding when the proteins were combined, but it is not clear whether the effects are additive or synergistic. As discussed in the manuscript, recruitment of MCM8-9 by HROB is a possibility, also supported by previous cellular data, which may not be very apparent in the reconstituted system. However, it is clear that the effect of HROB extends well above any potential recruitment function.

Figure R3: Electrophoretic mobility shift assay with MCM8-9 in the presence or absence of HROB using a HJ at no added (top) and 100 mM NaCl (bottom) with 3 mM EDTA. The red asterisk indicates the position of the radioactive label. Panel shows representative experiments.

Reviewer #3 (Remarks to the Author):

In this study, the authors employed multiple approaches to investigate the function of MCM8-9 helicase and its regulation by HROB. Molecular modeling was used to identify key residues from both HROB and MCM8-9 that are essential for HROB binding to MCM8-9. The authors showed that these interactions are crucial for stimulating MCM8-9's helicase activity. The authors also proposed that HROB binding activates the ATPase activity of MCM8-9, although the evidence for this claim is limited. The study further examined the activity of MCM8-9 on various DNA substrates and discovered that the MCM8-9 helicase favors Holliday Junction (HJ) and Y-shaped DNA templates. Overall, these findings offer novel insights into the function of MCM8-9 helicase, which would be of interest for the fields of DNA replication, damage repair and related areas. However, despite the reviewer's enthusiasm for this work, the present manuscript has various issues and concerns that need addressing.

Major concerns:

1. The authors need to provide concrete evidence to substantiate their claim in the manuscript's title that MCM8-9 operates as a hexamer for DNA unwinding through forming a complex with HROB.

RESPONSE: In the previous version of the manuscript, we have used

- (a) Disruption of the complex between MCM8-9 and HROB abrogates DNA unwinding
- (b) EMSA assays to show that ATP binding locks a MCM8-9 ring on ssDNA
- (c) AlphaFold2 modeling and mass photometry showing that MCM8-9 form hexamers, from dimer building blocks
- (d) Helicase assays showing that the ATPase site, which is reconstituted from subunits forming oligomer species higher than a dimer, is necessary for DNA unwinding (a conclusion that is further supported by the interface mutants).

These data, together with the in the meantime published structural studies (PMID 37535404 and 37309874) strongly suggest that MCM8-9 functions as a hexamer in DNA unwinding. However, a direct and completely unambiguous demonstration is very tricky. We will therefore remove the "hexamer" from the title, and discuss the conclusions with possible caveats in the text.

Nevertheless, in the revised version of the manuscript, we additionally

- (a) better demonstrate that HROB binds MCM8-9 stoichiometrically across the stable interface, and hence does not interfere with its hexameric structure (new Fig. 1a-c, and 1i and 6f).
- (b) present mass photometry experiments showing that even the dimeric MCM8 (R309D)-9 mutant forms hexamers only when DNA and ATP are present, which further supports our conclusions (new Supplementary Fig. 9c-f).

2. The authors found that the OB domain of HROB plays an important role in stimulating the helicase activity of MCM8-9. However, It is not clear how the removal of the OB domain affects the affinity of HROB binding to MCM8-9. This should be addressed.

RESPONSE: Our AlphaFold2 modeling identified a potential interaction between the OB fold domain of HROB and MCM9 (Fig. 1i). We have designed point mutants affecting either side of the interface, and show that the mutants disrupt the unwinding function of the ensemble (Fig. 1m), and also HROB binding to MCM8-9 both in cells (Fig. 1l) and *in vitro* (new Supplementary Fig. 2a,b).

In the revised manuscript, we show that the removal of the entire OB fold disrupts the interaction with MCM8-9 similarly as by the identified point mutations (new Supplementary Fig. 1g and 2a,b). We also better demonstrate that the deletion of the OB fold domain instead does not affect DNA binding of HROB (new Supplementary Fig. 1e,f).

3. The authors used molecular modeling to predict the structures of MCM8-9 hexamer with and without HROB. However, since the structures of related MCM8-9 structures, including a full-length chicken MCM8-9 (gMCM8-9, PDB: 7W7P; EMD-32346) and the NTD ring of human MCM8-9 (hMCM8-9) with and without HROB bound (Li et al 2021, Structure; Weng et al 2023, eLife) have been determined, it is important to compare the predicted structures with the available real ones to assess the reliability of the predictions.

RESPONSE: Please see our response to Reviewer 1.

4. The recent structure of hMCM8-9-NTD bound with HROB (Weng et al 2023, eLife) revealed conformational changes in the NTD ring of hMCM8-9; however, the HROB is not visible. This observation suggests that the flexible HROB may have multiple weak and/or transient binding surfaces on MCM8-9, similar to the structure of CMG bound with Mcm10 (Mayle et al 2018 PNAS; Yuan et al 2020 Nat Commun). Therefore, the authors should validate the predicted structure. Concrete evidence should be provided to demonstrate whether HROB binds to MCM8-9 hexamer. If the assembly is unstable, crosslinking should be applied for fixation. The relevant MCM8-9-HROB should be isolated for structure determination using cryo-EM and cross-linking mass spectrometry analysis. These analyses will provide concrete structural information, elevating the quality of this paper to a level suitable for publication in Nature Communications.

RESPONSE: We now demonstrate that HROB binds to the MCM8-9 dimer with 1:1 stoichiometry (new Fig. 1c). As noted in response to Reviewer 1 (comment 1), the interaction between MCM8-9 and HROB is however rather weak.

To further address the reviewer's comment, we have immobilized HROB, and performed pulldown experiments with MCM8, MCM9, and MCM8-9 complexes, present either a dimer (MCM8 (R309D)-9 mutant) or the wild type complex consisting of both dimers and hexamers. The strongest interaction was observed with the two MCM8-9 complexes (both dimer and dimer/hexamers interacted in the same way), followed by MCM9 alone, while MCM8 alone was not detected on the silver-stained gel (new Supplementary Fig. 9b). These experiments are in full agreement with our AlphaFold2 model that places HROB across the stable heterodimer interface (termed interface I in our study) of MCM8-9 (Fig. 1i and 6f).

We would like to kindly ask the reviewer to consider the requests for cryo-EM structure determination above the scope of this study, especially as it appears to be experimentally challenging even for structural labs.

5. The authors claim that HROB-F553E mutant loses the interaction with MCM8-9. This mutation site is in the OB domain of HROB. However, previous results from the same research team showed that the OB domain of HROB is dispensable for its binding to MCM8-9 (Huang et al 2020 Nat Commun). HROB-1-432 mutant, in which the OB domain is removed, can pull down MCM8-9. Therefore, it is unclear how a single point mutation, F553E, completely abolish the binding of HROB to MCM8-9 as shown in Extended Fig. 1c. This needs to be addressed.

Figure R4: Panels from Huang et al 2020, showing the interaction of HROB (termed MCM8IP) and MCM8-9 in cell extracts.

RESPONSE: The Huang et al. paper (Figure R4) shows the highest interaction of MCM8-9 with full-length HROB, and with a variant lacking the first 395 residues, showing that the interaction occurs primarily through HROB residues 396 to the end of the protein. The analysis then identifies a region between residues 414 and 432, as well as a region downstream of 432 that are both important for the interaction. Therefore, mutants lacking the OB fold domain were found to be notably impaired in the interaction in the Huang et al. paper. We have referred to it somewhat incorrectly and now clarified this point in the text.

Our analysis is qualitatively in a good agreement with Huang et al: one of the identified interaction sites maps to HROB around residue 405 (with MCM8), and the other one around residue 553 (within the OB fold domain, with MCM9). Also in our case, we can detect residual interaction between MCM8-9 and HROB lacking the OB fold domain (new Supplementary Fig. 1g). There are some quantitative differences; however, these are not unexpected, as we cannot directly compare affinities based on pulldowns from cell extracts (truncation mutants can be misfolded, proteins post-translationally modified, competition with other proteins etc.) with those carried out with recombinant proteins and their point mutants *in vitro*.

6. The authors found that ATP hydrolysis by MCM8-9 is only slightly enhanced by HROB. However, HROB induces a significant increase in the activity of MCM8-9 in DNA unwinding. It is important to address this observation to establish a possible mechanism by which HROB regulates the function of MCM8-9 in DNA unwinding.

RESPONSE: We believe that the observation reflects the capacity of MCM8-9 to hydrolyze DNA non-productively (without DNA unwinding), which was observed with many other motor proteins. We speculate in Discussion: "HROB may facilitate productive ATP hydrolysis, possibly by coordinating the conformational changes (within MCM8-9) with DNA translocation and unwinding."

Importantly, while the MCM8-9 hexamer AF2 model is rather symmetrical, it has been proposed that functional MCM helicases use an asymmetrical rotating mechanism to unwind DNA (as reviewed in PMID: 37244171), with the different subunits binding either ATP or ADP molecules asymmetrically. Consequently, an asymmetrical rather than symmetrical arrangement of the rings could better account for the functional states of the MCM8-9 helicase. The HROB bound-state, by restricting the distance between the N-ter and C-ter domains of MCM9 (Fig. 6f), could help trigger this asymmetry. A deformation of the N-ter ring has been reported upon HROB binding together with significant rotations between the N-terminal domains (eLife, PMID: 37535404), although neither the HROB density nor the conformation of the C-ter ring could be resolved in that case. Such a structural perturbation could be fully in line with the induced rotation observed in the AF2 models upon HROB binding.

7. In addition, despite the presence of HROB, the authors observed low processivity of MCM8-9 in DNA unwinding. The authors should discuss how this property of MCM8-9 might be applied in homologous recombination?

RESPONSE: This a question that we have been asking ourselves, and the honest answer is that we do not know. We speculate that MCM8-9 may be involved in quality control of homologous recombination, possibly disrupting aberrant joint molecules. MCM8-9 was proposed to promote recombination associated DNA synthesis, and the very low processivity might not be compatible with that model. However, there may be additional parameters (protein co-factors, post-translational modifications etc.) that could change MCM8-9 behavior. These are all unsubstantiated speculations, and interesting questions for future research.

REVIEWER COMMENTS

Reviewer #1 (Remarks to the Author):

I appreciate the extensive revision that the authors have undertaken to make this manuscript better. However, more questions are raised with this revision because of the new data. Without a cryoEM structure of the hexameric MCM8/MCM9 bound to DNA, the molecular modelling and biochemical evidence are not convincing enough for the authors to make the case that the MCM8/9 hexamer is the active physiological form of the helicase. For example, without the actual cryoEM structure of MCM8/9 hexamer bound to its substrate DNA, it is not clear whether one, two or three HROB is associated with the hexamer. Is the central channel large enough for ssDNA to pass through? The finding that mutants (R309D and D230R) that disrupt interface II and inhibit hexamer formation but can form dimers have higher DNA unwinding activities than the wild-type dimer or hexamer goes against the argument that the active form of the helicase is a hexamer. Comparing the alphafold2 predicted ring model with the published human ADP bound MCM8-9 structure, neither of which contain DNA, and then speculate on the conformational changes of the helicase during DNA unwinding does not lend confidence to their speculation. The study of the structure and function of MCM8/9 helicase has matured to a point that concrete evidence is required to move forward. The physical size that the authors use as evidence to support the hexamer is the active form could be explained by the unwinding activities coming from dimers disintegrated from the hexamers in the assay. All in all, I think this MS belongs to a more specialized journal and does not meet the interest of the readers of Nature Comm.

Reviewer #2 (Remarks to the Author):

In this revised manuscript, the authors addressed all of my criticisms and I have been convinced by their explanation and the new data. This work provides important information on the mechanism of MCM8-9 helicase activation. I think this work is now suitable for publication in Nature Communications.

Reviewer #3 (Remarks to the Author):

In this revised manuscript, the authors investigated the potential role of HROB in modulating the activities of MCM8-9 helicase in response to DNA damage. While the authors have made additional analyses and addressed some concerns raised by this reviewer. However, there are still major concerns that have not been adequately addressed. The authors have not provided solid evidence to support their conclusion that MCM8-9 operates as a hexamer for DNA unwinding and the precise role of HROB in this process remains unclear at a molecular level. As a result, how HROB stimulates the activities of MCM8-9 in DNA unwinding and remains enigmatic. Moreover, the biochemical results presented are contradictory to those from the single molecule analysis. Addressing these major concerns and incorporating these additional experiments would greatly strengthen the manuscript and provide novel mechanistic insights. Without these improvements, the impact of the manuscript remains incremental, and it may be more suitable for a specialized journal.

Suggestions:

1. The authors should isolate the MCM8-9-HROB in complex with appropriate DNA templates and perform negative staining EM imaging analysis. 2D averages should be conducted to visualize the stoichiometry of the MCM8-9-HROB during DNA unwinding. Apply crosslinking if necessary.
2. The authors should perform crosslinking mass spectrometry with the MCM8-9-HROB samples to examine the binding surfaces between HROB and MCM8-9.

Reviewer #1 (Remarks to the Author)

I appreciate the extensive revision that the authors have undertaken to make this manuscript better. However, more questions are raised with this revision because of the new data. Without a cryoEM structure of the hexameric MCM8/MCM9 bound to DNA, the molecular modelling and biochemical evidence are not convincing enough for the authors to make the case that the MCM8/9 hexamer is the active physiological form of the helicase. For example, without the actual cryoEM structure of MCM8/9 hexamer bound to its substrate DNA, it is not clear whether one, two or three HROB is associated with the hexamer. Is the central channel large enough for ssDNA to pass through? The finding that mutants (R309D and D230R) that disrupt interface II and inhibit hexamer formation but can form dimers have higher DNA unwinding activities than the wild-type dimer or hexamer goes against the argument that the active form of the helicase is a hexamer.

REPLY: We are not sure which new data the reviewer refers to. The data on R309D and D230R mutants were already included in the first version of the manuscript, and we have provided further evidence suggesting that this does not go against the model that MCM8-9 functions as a hexamer. The mutants helped to delineate the stable interface I (between MCM8 and 9 subunits in our cartoons) from unstable interface II (impacted by the mutations, between MCM9 and MCM8 in our cartoons).

While on first sight the data might indicate what the reviewer notes, our follow up experiments clearly support the notion that MCM8-9 dimers do not support unwinding, and MCM8-9 functions as a hexamer to unwind DNA.

(1) The R309D and D230R mutations weaken interface II, reducing non-productive hexamers in solution that assemble before loading on DNA. We show that ATP hydrolysis at this interface II (which only assembles when dimers associate to higher order complexes), even in the R309D and D230R hyperactive mutants, is essential for DNA unwinding, unlike the ATPase site at interface I (Fig. 6E). These data very clearly demonstrate that MCM8-9 dimers (having ATPase only at interface I) cannot unwind DNA.

(2) We noted that hexamers assembled in solution without DNA are not active, MCM8-9 needs to assemble from dimers and form hexameric rings on DNA, in the presence of ATP, as shown with various mutants and topological forms of DNA (Fig. 5). While the R309D and D230R mutants are dimers in solution, data in Fig. S9 using mass photometry indicate that they form hexamers on DNA in the presence of ATP, most likely during DNA unwinding.

Comparing the alphafold2 predicted ring model with the published human ADP bound MCM8-9 structure, neither of which contain DNA, and then speculate on the conformational changes of the helicase during DNA unwinding does not lend confidence to their speculation. The study of the structure and function of MCM8/9 helicase has matured to a point that concrete evidence is required to move forward. The physical size that the authors use as evidence to support the hexamer is the active form could be explained by the unwinding activities coming from dimers disintegrated from the hexamers in the assay. All in all, I think this MS belongs to a more specialized journal and does not meet the interest of the readers of Nature Comm.

REPLY: The speculation on the conformational changes is only included in the discussion, and is clearly indicated as such "We speculate that repetitive structural changes in the relative orientation of the MCM8-9 N- and C-terminal rings induced by HROB". Our reasoning supporting the notion that MCM8-9 functions as a hexamer is described in the third paragraph of Discussion, with references to specific Figures. The comment was added in response to Reviewer 3: "It is important ... to establish a possible mechanism by which HROB regulates the function of MCM8-9 in DNA unwinding".

Beyond the biochemical data presented here, we note that all MCM homologues, from all kingdoms of life, unwind DNA as hexameric complexes (formed from single, double or 6 different polypeptides). There is sufficient space to accommodate ssDNA in the opening of the hexameric ring. The opening of MCM8-9 pore in the model as a diameter of 25 Å, which matches that of the MCM complex of archaea (DNA free): (PDB_4R7Y) = 24.7 Å. The size of the pore gets reduced upon DN binding: MCM2-7 with ssDNA (PDB_6XTX) = 15.5 Å. The architecture of the MCM family including MCM8-9 is sufficiently conserved to map most of the key residues involved in ssDNA contacts in the opening (key DNA binding residues: e.g. MCM8_K493 = MCM9_391 = MCM6_R435 = MCM4_K549 = MCM7_R420 = MCM5_R420; MCM8_K543 = MCM9_K440 = MCM6_K486 = MCM4_K600 = MCM7_K471 = MCM3_K480 = MCM5_K471).

Reviewer #2 (Remarks to the Author)

In this revised manuscript, the authors addressed all of my criticisms and I have been convinced by their explanation and the new data. This work provides important information on the mechanism of MCM8-9 helicase

activation. I think this work is now suitable for publication in Nature Communications.

REPLY: Thank you.

Reviewer #3 (Remarks to the Author)

In this revised manuscript, the authors investigated the potential role of HROB in modulating the activities of MCM8-9 helicase in response to DNA damage. While the authors have made additional analyses and addressed some concerns raised by this reviewer. However, there are still major concerns that have not been adequately addressed. The authors have not provided solid evidence to support their conclusion that MCM8-9 operates as a hexamer for DNA unwinding and the precise role of HROB in this process remains unclear at a molecular level. As a result, how HROB stimulates the activities of MCM8-9 in DNA unwinding and remains enigmatic.

REPLY: We show that HROB interacts with both MCM8 and MCM9 subunits (and not only with MCM8 as thought previously). Interaction with both subunits is essential for DNA unwinding, as demonstrated by reciprocal mutagenesis. Also, we show that HROB stimulates MCM8-9 downstream of its assembly on ssDNA (previously, it was thought that HROB is a recruitment factor, which we clearly show that is not sufficient). The AlphaFold2 model and data from PMID: 37535404 are suggestive of structural changes in MCM8-9 upon HROB binding, consistent with structural transitions required for DNA unwinding as observed for other members of the MCM family in diverse kingdoms of life. The molecular details of the model are obviously speculative (as indicated in Discussion); however, we believe that our data brought the field forward and reveal unexpected properties of HROB stimulation of MCM8-9.

Moreover, the biochemical results presented are contradictory to those from the single molecule analysis.

REPLY: We are very puzzled by this comment. We are not aware of any contradiction between the biochemical and single molecule data: both indicate that HROB is necessary for DNA unwinding, the complex prefers to act on branched DNA structures, and does not have processive DNA unwinding activity. The only case where the data may not seem to match perfectly relates to experiments where we observed that a large excess of HROB is inhibitory (in bulk gel-based assays). Using an unrelated helicase (Bloom, BLM), we show that "the inhibition of MCM8-9 by high HROB concentrations is non-specific, stemming likely from a competition for DNA". It therefore has nothing to do with the stoichiometries of MCM8-9 and HROB, as demonstrated by the single molecule experiments (Fig. S6). The non-specific inhibition is not observed in the single molecule setup due to the low DNA concentration that is below the binding constant.

Addressing these major concerns and incorporating these additional experiments would greatly strengthen the manuscript and provide novel mechanistic insights. Without these improvements, the impact of the manuscript remains incremental, and it may be more suitable for a specialized journal.

Suggestions:

1. The authors should isolate the MCM8-9-HROB in complex with appropriate DNA templates and perform negative staining EM imaging analysis. 2D averages should be conducted to visualize the stoichiometry of the MCM8-9-HROB during DNA unwinding. Apply crosslinking if necessary.
2. The authors should perform crosslinking mass spectrometry with the MCM8-9-HROB samples to examine the binding surfaces between HROB and MCM8-9.

REPLY:

The reviewer asks for EM structure of the complex to define the (1) stoichiometry of MCM8-9 and HROB and (2) the interaction interfaces. We define the stoichiometry using mass photometry (real-time single molecule assay) to show that one HROB molecule binds per one MCM8-9 complex. Our mass photometry and reciprocal pulldown experiments also indicate that the interaction is very transient. (2) We verify the interaction sites predicted by the AlphaFold2 model by reciprocal mutagenesis (both on MCM8-9 and on HROB sides) and subsequent analysis of the recombinant mutants *in vitro* and also, upon expression in cells, which functionally validated the model. By designing the separation of function mutants, we went beyond a simple description of the interfaces (Fig. 1).

We opted not to pursue cryoEM analysis as there are already two studies that attempted that (PMID 37535404, 37309874). None of the structures could not resolve HROB or DNA, most likely because the interactions are transient and the complexes are non-homogeneous. We would certainly encounter similar limitations. Even if we were able to resolve the complexes, possibly upon crosslinking (prone to artefacts), we could not conclude that they represent functional species active in DNA unwinding, as opposed to non-productive complexes (see above), certainly not within the scope of this study. Hence, we opted for a modeling approach, supported by

mutagenesis and biochemistry, which provides us with a direct functional validation.

We believe the traditional EM-based approach is not suitable in the case of dynamic systems such as MCM8-9 with HROB, and may not be particularly helpful for mechanistic understanding of enzymatic activities (PMID 37535404, 37309874). Nevertheless, we believe that the points raised by reviewer are supported by complementary approaches in our manuscript.

REVIEWER COMMENTS

Reviewer #4 (Remarks to the Author):

Review of "Mechanism of DNA unwinding by MCM8-9 in complex with HROB" by Cejka and colleagues for Nat. Comm.

1) Regarding the issue of the hexameric structure as the active form assembled from dimers of trimers, the authors have done a reasonable job showing this is the case and based upon all other MCM complexes, the hexamer is the active form, but MCM8/9 is different for sure. Their R309D and D230R mutants are particularly informative; it was surprising that these mutant more stable heterodimers are more active for unwinding than the hexamer. The author's response that these mutants combined with ATPase mutants do not support unwinding (Fig 6e) as evidence for a hexamer is convincing from a biochemical perspective, but this is not unambiguous proof that the hexamer is the active form. Also, does that favor the idea that there must be a MCM8/9 hexamer loader as for all the other MCMs?

2) Have the authors shown that the hexamer is in a dynamic or stable equilibria? For example, if they isolate the hexameric species, then wait a while and do mass photometry again, are there dimer and hexamer populations? Or can they take the hexamer form and preincubate with DNA substrate at different times (10-60 min for example) in the absence of ATP before initiating unwinding with ATP to see if the hexamer can reequilibrate into dimers and assemble onto DNA that way?

3) Reviewer 1 is correct in that the authors cannot delineate between 1, 2, or 3 HROB molecules binding to the MCM8/9 hexamer and that cryoEM structure is needed to show that; however, even with that, it won't be clear biochemically which stoichiometric state is the active form. Fluorescent labelling experiences combined with unwinding would be needed to show that and would be outside the scope here I would think.

4) Regarding the stimulation of unwinding by HROB. It is clear that HROB stimulates MCM8/9 unwinding, however the mechanism by which it does so is still not yet established. They do show that HROB likely acts after MCM8/9 binding to DNA, as MCM8/9 can load onto circular DNA without nucleotide or HROB, but what happens at that point is not clear. I think it is fine to speculate on the mechanism based on their data, but they are using the AlphaFold model in Fig 6F, which is the dimer MCM8/9 instead of any hexamer model with HROB, and therefore also speculative. And so, they are using speculative data from a dimer MCM8/9-HROB model to speculate on a mechanism of unwinding for a hexamer MCM8/9-HROB complex. That said, the flexible C-terminal domain of MCM8/9 shown in PMID 37535404 & 37309874 seems to limit productive unwinding, and if binding of HROB locks in a more productive conformation of the CTD compared to the NTD, then that would be an important advance. However, the last sentence in the abstract is purely speculative and may not be appropriate to include there. Might be better to leave that for the Discussion.

5) It appears that the magnetic tweezer experiments do not support unwinding a fork DNA substrate (Supp Fig 5 b-c) but the bulk experiments do (Fig 2). A better explanation may be needed to reconcile this difference. In Suppl Fig 5b-c they used a 5' arm in their tweezers setup. Wondering if loading is inhibited there? Have they tried a 3' arm instead?

RESPONSE TO REFEREE'S COMMENTS

1) Regarding the issue of the hexameric structure as the active form assembled from dimers of trimers, the authors have done a reasonable job showing this is the case and based upon all other MCM complexes, the hexamer is the active form, but MCM8/9 is different for sure. Their R309D and D230R mutants are particularly informative; it was surprising that these mutant more stable heterodimers are more active for unwinding than the hexamer. The author's response that these mutants combined with ATPase mutants do not support unwinding (Fig 6e) as evidence for a hexamer is convincing from a biochemical perspective, but this is not unambiguous proof that the hexamer is the active form. Also, does that favor the idea that there must be a MCM8/9 hexamer loader as for all the other MCMs?

We are pleased that the reviewer found the experiments with the R309D and D230R mutants informative, and that the reviewer agrees that the data support evidence for a hexamer from a biochemical perspective. The conclusion that a hexamer is the active species is additionally supported by evolutionary conservation of residues involved in the contact with DNA, which are positioned in the central channel, between hexameric MCM2-7 and the MCM8-9 model (consistent with the published CryoEM structures of hexameric MCM8-9) (Reviewer Figure 1). Key DNA binding residues: e.g. MCM8_K493 = MCM9_391 = MCM6_R435 = MCM4_K549 = MCM7_R420 = MCM5_R420; MCM8_K543 = MCM9_K440 = MCM6_K486 = MCM4_K600 = MCM7_K471 = MCM3_K480 = MCM5_K471).

Reviewer Figure 1: Depiction of key conserved residues involved in contact with DNA between the AF2 model MCM8-9 and MCM2-7 helicases.

Nevertheless, we added to the text including the Abstract phrases such as "*Biochemical experiments demonstrate*" and "*In vitro, ...*", to clarify that the conclusions are based on biochemical experiments. Finally, we also added a comment in the Discussion on the limitation of our study "*Although our biochemical experiments and available CryoEM structures^{20,23} point towards MCM8-9 operating as a hexamer, the data are not entirely unambiguous that hexamer is the active form under all conditions.*" It is well possible that a MCM8-9 loader exists that will facilitate the formation of a productive helicase complex, however this does not appear to be HROB.

2) Have the authors shown that the hexamer is in a dynamic or stable equilibria? For example, if they isolate the hexameric species, then wait a while and do mass photometry again, are there dimer and hexamer populations? Or can they take the hexamer form and preincubate with DNA substrate at different times (10-60 min for example) in the absence of ATP before initiating unwinding with ATP to see if the hexamer can reequilibrate into dimers and assemble onto DNA that way?

We have repeated the size exclusion chromatography (Reviewer Figure 2 panel a-b). We observed that the hexamer is unstable, and a major part of it falls apart or aggregates during a 2 h incubation period

(panel c). Instead, the dimer-rich sample was stable (panel d). We also noted that the hexamer-rich sample tends to aggregate on DNA in electrophoretic mobility shift assays (species get stuck in wells of agarose gels). These data fully agree with the diminished DNA unwinding capacity previously reported in the manuscript. We have further commented on this point:

"Furthermore, the hexameric complex was unstable and prone to aggregation showing that the MCM8-9 hexamers that assembled in solution without the DNA substrate may be inactive complexes"

Reviewer Figure 2. Analysis of MCM8-9 oligomeric species.

Panel a and b: MCM8-9 preparation was subject to size exclusion chromatography and separated by size (panels a and b).

Panel c: Hexamer-rich sample was assayed by mass photometry at defined time-points after size exclusion chromatography. We observed the fraction of the hexamers to diminish over time, and we observed increased fractions of MCM8 and MCM9 proteins alone, as well as protein aggregates (not shown).

Panel d: Dimer-rich sample was assayed by mass photometry at defined time-points after size exclusion chromatography. The sample was stable.

Panel e: Hexamer-rich sample shows signs of aggregation when incubated with DNA (well-shifts). The MCM8-9 rings assembled from dimers on DNA (see also Fig. 5) instead migrated into the gel.

3) Reviewer 1 is correct in that the authors cannot delineate between 1, 2, or 3 HROB molecules binding to the MCM8/9 hexamer and that cryoEM structure is needed to show that; however, even with that, it won't be clear biochemically which stoichiometric state is the active form. Fluorescent labelling experiments combined with unwinding would be needed to show that and would be outside the scope here I would think.

We are clarifying this point in the Discussion: *"We could detect stoichiometric binding of HROB to MCM8-9 heterodimer, but we could not obtain experimental evidence for how many HROB molecules can simultaneously bind to a MCM8-9 hexamer."*

4) Regarding the stimulation of unwinding by HROB. It is clear that HROB stimulates MCM8/9 unwinding, however the mechanism by which it does so is still not yet established. They do show that HROB likely acts after MCM8/9 binding to DNA, as MCM8/9 can load onto circular DNA without nucleotide or HROB, but what happens at that point is not clear. I think it is fine to speculate on the

mechanism based on their data, but they are using the AlphaFold model in Fig 6F, which is the dimer MCM8/9 instead of any hexamer model with HROB, and therefore also speculative. And so, they are using speculative data from a dimer MCM8/9-HROB model to speculate on a mechanism of unwinding for a hexamer MCM8/9-HROB complex. That said, the flexible C-terminal domain of MCM8/9 shown in PMID 37535404 & 37309874 seems to limit productive unwinding, and if binding of HROB locks in a more productive conformation of the CTD compared to the NTD, then that would be an important advance. However, the last sentence in the abstract is purely speculative and may not be appropriate to include there. Might be better to leave that for the Discussion.

We agree that the last part of the abstract "possibly coordinating ATP hydrolysis with structural transitions accompanying translocation of MCM8-9 on DNA" is a speculation, and we removed it from the Abstract and the end of Introduction. As the reviewer agrees, our data show that MCM8-9 loads on DNA without HROB, and HROB stimulates after MCM8-9 binding to DNA. We introduce the speculative model only at the end of Discussion, with the following preamble: "*How can HROB stimulate DNA unwinding by MCM8-9 downstream of its assembly on DNA? Our biochemical data do not provide direct evidence to answer this question, but the AlphaFold2 model together with the recent cryoEM study of the chicken complex²³ allow us to propose a hypothetical model.*"

5) It appears that the magnetic tweezer experiments do not support unwinding a fork DNA substrate (Supp Fig 5 b-c) but the bulk experiments do (Fig 2). A better explanation may be needed to reconcile this difference. In Suppl Fig 5b-c they used a 5' arm in their tweezers setup. Wondering if loading is inhibited there? Have they tried a 3' arm instead?

In the magnetic tweezers experiments on the fork substrate we probe the difference between the extension of ssDNA and dsDNA at the applied force. From the molecular dimensions, the maximum extension difference is $0.51 \text{ nm} - 0.34 \text{ nm} = 0.17 \text{ nm}$ per base pair (about 1/6 bp). Thus, an unwinding of 30 bp as probed in the bulk assays would correspond in the tweezers assay only to a length difference of max 5 nm. For the given molecule length, this is within the measurement noise particularly when probing a transient, few second-lived event. For the Holliday junction substrate this is different. Here, the unwinding of one horizontal branch by 1 bp increases the DNA length of the two vertical branches by 2 bp, leading to about 1 nm length change per base pair. An unwinding of 30 bp corresponds in this case to about 30 nm in length, which is easily detectable. Thus, we expect limited unwinding activity also on our fork construct but are unable to detect it because of the limited processivity of MCM8-9. [We also note that a free 3' end was not necessary in the experiments with M13-based DNA, so the tweezer experiments with the 3' end are unlikely to be informative].

Overall we conclude that MCM8-9 has only very limited processivity (supported by bulk experiments in Fig. 3a and the magnetic tweezer experiments with the HJ substrate in Fig. 3b-d). Because the fork substrate is not suitable for measuring limited unwinding processivity and hence essentially reports negative data, and because this created a confusion during the review process, we opted to remove the fork substrate tweezer experiments from the manuscript (Supplementary Fig. 5bc).

REVIEWERS' COMMENTS

Reviewer #3 (Remarks to the Author):

The concerns I previously raised (see below) have not been addressed by the authors. While the manuscript suggests a potential role of MCM8/9 in DNA unwinding, with the assistance of HROB, the underlying mechanism has not been elucidated. Whether MCM8/9 functions as a hexamer or not is not solved. The biochemical results presented are contradictory to those from the single molecule analysis. As it stands, the manuscript falls short of the standard expected for publication in Nature Communications. It is more appropriate for submission to a specialized journal.

Previous comments:

In this revised manuscript, the authors investigated the potential role of HROB in modulating the activities of MCM8-9 helicase in response to DNA damage. While the authors have made additional analyses and addressed some concerns raised by this reviewer. However, there are still major concerns that have not been adequately addressed. The authors have not provided solid evidence to support their conclusion that MCM8-9 operates as a hexamer for DNA unwinding and the precise role of HROB in this process remains unclear at a molecular level. As a result, how HROB stimulates the activities of MCM8-9 in DNA unwinding and remains enigmatic. Moreover, the biochemical results presented are contradictory to those from the single molecule analysis. Addressing these major concerns and incorporating these additional experiments would greatly strengthen the manuscript and provide novel mechanistic insights. Without these improvements, the impact of the manuscript remains incremental, and it may be more suitable for a specialized journal.

Suggestions:

1. The authors should isolate the MCM8-9-HROB in complex with appropriate DNA templates and perform negative staining EM imaging analysis. 2D averages should be conducted to visualize the stoichiometry of the MCM8-9-HROB during DNA unwinding. Apply crosslinking if necessary.
2. The authors should perform crosslinking mass spectrometry with the MCM8-9-HROB samples to examine the binding surfaces between HROB and MCM8-9.

Reviewer #4 (Remarks to the Author):

I think the authors have made every attempt to reconcile differences with the reviewers and have added, updated, and removed data as discussed back and forth throughout several reviews. Therefore, the data, models, and interpretations are now plausible and suitable for publication.